# Gene activation guided by nascent RNA-bound transcription factors

Ying Liang[1,2,10], Haiyue Xu[1,2,10], Tao Cheng [3], Yujuan Fu[1], Hanwei Huang[1], Wenchang Qian[4], Junyan Wang[1], Yuenan Zhou [5], Pengxu Qian [4], Yafei Yin [5], Pengfei Xu[3], Wei Zou [6,7] ✉ & Baohui Chen [1,2,8,9] ✉

Technologies for gene activation are valuable tools for the study of gene functions and have a wide range of potential applications in bioengineering and medicine. In contrast to existing methods based on recruiting transcriptional modulators via DNA-binding proteins, we developed a strategy termed Narta (nascent RNA-guided transcriptional activation) to achieve gene activation by recruiting artificial transcription factors (aTFs) to transcription sites through nascent RNAs of the target gene. Using Narta, we demonstrate robust activation of a broad range of exogenous and endogenous genes in various cell types, including zebrafish embryos, mouse and human cells. Importantly, the activation is reversible, tunable and specific. Moreover, Narta provides better activation potency of some expressed genes than CRISPRa and, when used in combination with CRISPRa, has an enhancing effect on gene activation. Quantitative imaging illustrated that nascent RNA-directed aTFs could induce the high-density assembly of coactivators at transcription sites, which may explain the larger transcriptional burst size induced by Narta. Overall, our work expands the gene activation toolbox for biomedical research.

Regulation of gene expression requires that the transcriptional machinery be precisely and efficiently assembled at specific genomic loci[1–3]. Transcription factors (TFs) ensure this specificity by recognizing and binding to specific DNA sequences to modulate gene expression through their effector domains[4,5]. TFs typically consist of two subdomains, a DNA-binding domain (DBD) and an activation domain (AD) which interacts with coactivator complexes to modulate transcriptional level[6–8]. Based on this principle, the recently developed DNA-targeting platform, CRISPR-Cas9, has enabled the recruitment of artificial transcription factors (aTFs) to any specific genomic site to induce endogenous gene activation, termed CRISPR activation (CRISPRa)[9–13]. The ability of CRISPRa to activate target genes by using

single sgRNAs enables genome-wide transcriptional activation screens[14,15]. However, the use of multiple sgRNAs tiled across the target gene promoter can significantly improve CRISPRa efficiency, suggesting that recruiting as many TFs as possible may be crucial for activating gene expression with wider dynamic ranges[9–11,16]. Thus, a new gene activation strategy based on this principle may effectively activate genes that are inaccessible to current CRISPRa methods. This could be highly desirable for some biological processes, such as the direct conversion of cell types and industrial applications[17,18].

Enhancers are *cis*-regulatory elements (small segments of DNA) bound by TFs and other components of the transcription apparatus that modulate the expression of cell identity genes[19–22]. Super-

[1]Department of Cell Biology and Bone Marrow Transplantation Center of the First Affiliated Hospital, Zhejiang University School of Medicine, Hangzhou, China. [2]Liangzhu Laboratory, Zhejiang University Medical Center, Hangzhou, China. [3]Women's Hospital, Zhejiang University School of Medicine, Hangzhou, China. [4]Center of Stem Cell and Regenerative Medicine, Zhejiang University School of Medicine, Hangzhou, China. [5]Department of Cell Biology, Zhejiang University School of Medicine, Hangzhou, China. [6]The Fourth Affiliated Hospital, Zhejiang University School of Medicine, Yiwu, China. [7]Insititute of Translational Medicine, Zhejiang University, Hangzhou, China. [8]Institute of Hematology, Zhejiang University & Zhejiang Engineering Laboratory for Stem Cell and Immunotherapy, Hangzhou, China. [9]Zhejiang Provincial Key Laboratory of Genetic & Developmental Disorders, Hangzhou, China. [10]These authors contributed equally: Ying Liang, Haiyue Xu. ✉e-mail: zouwei@zju.edu.cn; baohuichen@zju.edu.cn

enhancers (SEs) are clusters of enhancers that are occupied by exceptionally high densities of interacting factors, including TFs, co-factors (e.g., p300, BRD4, and MED1), RNA polymerase II, and non-coding RNAs[23]. MED1 (also known as TRAP220) is a key subunit of the Mediator complex, which forms a bridge between the RNA polymerase II and transcriptional activators[24,25]. The coactivator p300 and its paralog CREB-binding protein (CBP) active transcription by facilitating transcriptional machinery assembly and by acetylating histones and other factors[26–28]. BRD4 is recruited by recognizing the acetylated lysine mediated by p300 and promotes transcriptional elongation of SE-associated pluripotency genes[26,29–31].

Active enhancers and super-enhancers can produce transcripts termed enhancer RNAs (eRNAs), which have been suggested to bring transcriptional activators to the promoters of neighboring protein-coding genes[32,33]. For example, eRNAs at enhancers can trap transcription factors such as yin and yang 1 (YY1) and enhance their binding to enhancers[34]. Furthermore, m6A-marked nascent RNAs (including pre-mRNAs and eRNAs) can recruit reader proteins to regulate transcription[35,36]. The density of TFs and co-factors assembled at SEs is estimated to be approximately tenfold the density of the same component at typical enhancers. SEs are therefore able to drive higher levels of transcription than typical enhancers and thus regulate genes with especially important roles in cell identity[37,38]. We reasoned that artificially concentrating high-density of transcriptional factors at transcription sites might be able to induce maximal activation of target genes. Motivated by this hypothesis, we developed a gene activation tool by repurposing nascent RNAs to recruit abundant aTFs at their transcription sites. Our method can activate a broad range of exogenous and endogenous promoters with high efficiency in various cell types.

## Results

### Design of Narta
Inspired by the principle of gene activation mediated by SEs, we hypothesized that the intron region in newly transcribed RNAs of a target gene could be repurposed to recruit artificial transcription factors (aTFs), which further recruit abundant coactivators at the transcription sites, leading to high levels of transcriptional activation. We therefore proposed a gene activation technology termed Narta (**Na**scent **R**NA-guided **t**ranscriptional **a**ctivation) (Fig. 1a). Toward this goal, we utilized our previous TriTag labeling system[39] and created a BFP^TriTag reporter driven by the mini-cytomegalovirus (miniCMV) promoter. The unique feature of the BFP^TriTag system is that BFP contains an intron harboring 12 copies of MS2 RNA hairpins, which can be selectively bound by synonymous tandem MS2 coat proteins (stdMCP)[40]. Thus, the fusion protein of stdMCP-tdTomato allows tracking of the nascent RNA production of TriTag-tagged genes in real time.

To recruit TFs through nascent RNAs produced by miniCMV-BFP^TriTag, we fused stdMCP to the transactivation domains of NF-κB p65 subunit (p65) and heat shock factor 1 (HSF1)[15,41] to generate stdMCP-p65-HSF1 (hereon referred to as stdMCP-PH). This fusion protein was expressed under the control of doxycycline (Dox)-inducible TRE3G promoter (Supplementary Fig. 1a). In addition, stdPCP-PH, which specifically recognizes PP7 but not MS2 hairpins[42], was also constructed as a negative control. To image Narta activation, we generated a reporter HeLa cell line (miniCMV-BFP^TriTag), in which the addition of Dox can induce stdMCP-PH expression. Additionally, nascent RNA production can be monitored by stdMCP-tdTomato, while protein expression levels can be quantified based on BFP imaging (Supplementary Fig. 1b). It is worth noting that the simultaneous use of stdMCP-tdTomato and stdMCP-PH to image and manipulate gene expression in the same cells may reduce the sensitivity of imaging and the efficiency of gene activation. An orthogonal activation or RNA reporter system would be an ideal design. However, our BFP^TriTag

harbors 12 copies of the MS2 sequence, which is still worth testing. When only the level of single-cell protein expression needs to be quantified for assessing Narta activation in the subsequent experiments, we used cells without stdMCP-tdTomato expression.

### Narta activates gene expression by modulating transcriptional bursts
We first examined whether nascent RNA-guided activators can induce transcriptional activation. In our system, MS2/stdMCP-tdTomato allows quantitative analysis of transcriptional bursting kinetics in real time. A number of studies have established the link of TF and transcriptional bursts[26,43,44]. It was previously suggested that TF concentration can modulate the burst frequency[43]. Thus, we explored whether Narta could activate transcription by altering transcriptional bursts. To this end, we monitored the transcriptional bursting of miniCMV promoter by imaging its production of nascent mRNAs labeled by stdMCP-tdTomato in HeLa cells. Real-time imaging revealed that nascent RNAs were produced in bursts (Supplementary Fig. 2a, b), which is consistent with previous studies[39,45,46]. To monitor Narta activation, Dox was added to induce stdMCP-PH expression for 12 h and then real-time imaging was recorded for 2 h. Quantitative analysis indicated that the addition of Dox induced more de novo RNA production (~6-fold) which was defined by the total intensity of individual stdMCP-tdTomato spots (Supplementary Fig. 2c). Moreover, the burst duration was increased by ~ 5-fold (-Dox: ~17 min; +Dox: ~85 min), whereas the pause duration remained nearly unchanged (-Dox: ~26 min; +Dox: ~27 min) (Supplementary Fig. 2d and Supplementary videos 1 and 2). Therefore, nascent RNA-guided TFs can modulate transcriptional bursts of target genes, resulting in increased gene activity.

We then confirmed the transcriptional activation capacity of Narta by performing real-time quantitative PCR (RT-qPCR). Following Narta activation for 48 h, the mRNA abundance of BFP was elevated by approximately eightfold (Supplementary Fig. 2e). We also quantified protein expression based on fluorescent imaging. The expression of BFP was dramatically increased by about 26-fold (Supplementary Fig. 2f). Together, these findings suggest that nascent RNA-guided TFs can activate transcription of the target gene from which the nascent RNAs are produced in human cells.

### Narta activates exogenous genes robustly in mammalian cells and zebrafish embryos
Having performed initial characterizations of Narta-mediated gene activation, we sought to assess its ability to activate more exogenous reporters in various cell types. miniCMV is widely reported as a weak promoter. Therefore, we further evaluated BFP^TriTag expression in HeLa cells driven by four stronger promoters, including the spleen focus-forming virus promoter (SFFV), the ubiquitous human cytomegalovirus promoter (CMV), the promoter of human elongation factor 1α (EF1α) and a hybrid CMV/β-actin promoter (CAG)[47]. Imaging analysis revealed that Narta activation resulted in significant increases in BFP protein expression driven by any of the promoters we tested (Supplementary Fig. 2g). Next, to test whether Narta works in other cell types, we examined Narta activation in Chinese hamster ovary (CHO) cells. BFP^TriTag expression levels driven by CAG and EF1α were quantified, respectively. The results indicated that Narta is as effective in CHO cells as it is in HeLa cells (Supplementary Fig. 2h). CHO cells are the predominant mammalian cell line used for producing high quantities of biotherapeutic proteins[48]. Thus, Narta may serve as a valuable strategy to boost the production of recombinant proteins in CHO cells.

We next tested whether nascent RNA-guided activators could induce gene expression in whole organisms. To do this, we designed and constructed a GFP tag for Narta activation in zebrafish, named GFP^Fish_NarTag, which harbors nine copies of MS2 repeats in its artificial intron (an intron from *pcfg1* gene) (Supplementary Fig. 3a). We

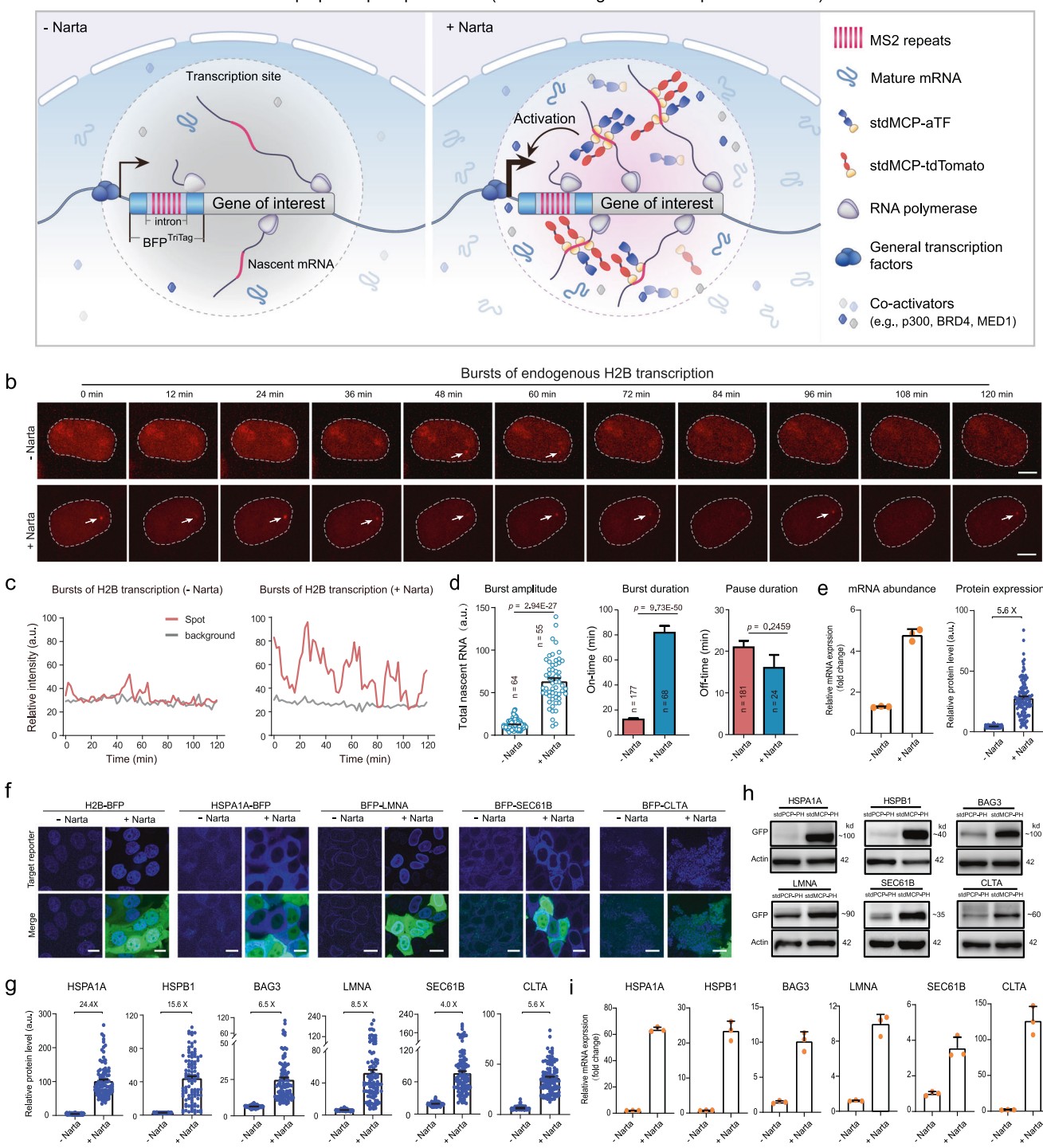

**Fig. 1 | Narta activates endogenous genes. a** Schematic depicting recruitment of artificial transcription factors (aTFs) or fluorescent reporters to nascent mRNAs by the MS2/MCP system. **b** Snapshots of representative HeLa cells showing the transcriptional bursting dynamics of H2B loci revealed by stdMCP-tdTomato without (top) or with (bottom) Narta activation. Scale bar, 5 μm. **c** Representative traces (red) of nascent transcripts produced at H2B loci from the cells in **b**. Gray traces illustrate the background signal in the nuclei. **d** Quantitative analysis of the burst amplitude (left), burst durations (middle) and pause durations (right) to show the bursting characteristics of H2B transcription. Burst amplitude is defined by the total intensity of individual stdMCP-tdTomato spots. *P* values were calculated by two-tailed Student's *t* test. **e** Quantifications of H2B-BFP transcription by qRT-PCR (*n* = 3 biological repeats) and protein expression (*n* = 100 cells) by fluorescent

imaging. **f** Representative images to show the activation of endogenous reporters by Narta. GFP indicates the successful transfection of stdMCP-PH. Scale bar, 10 μm. **g** Quantifications of protein expression level based on fluorescent imaging in **f**. Each dot represents a cell. *n* = 100 cells. **h** Measurement of target protein abundance by Western blotting. Endogenous genes were tagged with GFP^TriTag and thus their expression was detected by GFP antibody. stdMCP-PH vectors were transfected to induce Narta activation, while stdPCP-PH transfection serves as the negative control. Actin was detected as an internal reference. The experiment was repeated two times with similar results. **i** qRT-PCR analysis of the mRNA expression level of various endogenous genes without or with Narta activation, *n* = 3 biological replicates. Data in Fig. 1 are all shown as mean ± s.e.m. Source data are provided as a Source Data file.

microinjected the mixture of CMV-mCherry & CMV-GFP, CMV-mCherry & CMV-GFP[Fish_NarTag] & CMV-stdMCP-PH or CMV-mCherry& CMV-GFP[Fish_NarTag] & CMV-stdPCP-PH into zebrafish embryos of 1-cell stage. We then assessed GFP and mCherry expression by fluorescent imaging when the embryos were developed into the stage of 80 to 90% epiboly (Supplementary Fig. 3b). The constitutive mCherry expression vector was used as the internal control to assess the amount of DNA injected across different conditions. Our data illustrated that Narta activation could specifically induce higher levels of GFP[Fish_NarTag] expression compared to GFP or GFP [Fish_NarTag] without activation (Supplementary Fig. 3c, d). In conclusion, our results based on various transgene reporters reveal that Narta is an effective tool for triggering gene activation in various cell types.

### Activation of endogenous promoters by Narta

Next, to monitor the activation of an endogenous promoter, we inserted TriTag into the C-terminus of human H2B via CRISPR/Cas9-mediated homologous recombination in HeLa cells which stably expressed stdMCP-tdTomato. Consistent with our previous finding[39], the transcription of H2B loci occurred in discontinuous bursts (Fig. 1b and Supplementary Video 3). Quantitative analysis showed that Narta (12 h after stdMCP-PH transfection) induced larger transcriptional burst sizes of H2B, including longer burst durations (control: ~13 min; Narta: ~62 min) and larger burst amplitudes (relative RNA intensity: 12.5 vs. 65.0). However, the mean pause duration was not significantly affected by Narta activation (Fig. 1c, d and Supplementary Video 4). These results suggest that Narta activation induced the production of more H2B transcripts, which was further confirmed by qRT-PCR. Consistent with transcriptional activation, the expression of H2B-BFP was dramatically increased by approximately sixfold upon Narta treatment (Fig. 1e).

To assess the general applicability of Narta for gene activation, we created fourteen additional endogenous reporters through integrating BFP[TriTag] to the N- or C-terminal of endogenous genes by CRISPR-editing, including SEC61B, CYB5B, CLTA, LMNA, HPDL, BAG3, HSPA1A, HSPA1B, HSPB1, HSPB8, β-actin, VAPB, TOMM70A and CBX1 (Fig. 1f and Supplementary Fig. 4a). We transfected these cell lines with stdMCP-PH (+Narta) or stdPCP-PH (- Narta) and quantified protein expression by fluorescent imaging at 48 h after transfection. Quantitative analysis revealed that Narta displayed high levels of activation of most tested genes (e.g., HSPA1A: 24-fold; HSPB1: 16-fold; HSPB8: 10-fold; BAG3: 6-fold; LMNA: 9-fold; HPDL: 7-fold) (Fig. 1g and Supplementary Fig. 4b). Notably, Western blot assays detected Narta-mediated production of more endogenous fusion proteins with correct molecular sizes (Fig. 1h). Moreover, mRNA abundance of target genes was dramatically increased measured by qRT-PCR and single-molecule fluorescence in situ hybridization (smFISH), demonstrating that Narta can robustly modulate the transcriptional level at endogenous loci (Fig. 1i and Supplementary Fig. 5).

Next, to further explore the general trend we observed within human HeLa cells, we examined Narta activation in HEK293T cells. Four endogenous genes, including HSPA1A, HSPB1, LMNA and HPDL, were tagged with BFP[TriTag]. Consistent with the results in HeLa cells, all these four genes could be activated with high efficiency (Supplementary Fig. 6). To test whether Narta allows multiplexed activation of endogenous genes, we created a HeLa cell line stably co-expressing H2B-BFP[TriTag], GFP[TriTag] -SEC61B and mCherry[TriTag]-LMNA by sequential CRISPR knockin experiments. Transfecting this cell line with stdMCP-PH successfully activated all three genes simultaneously (H2B, 6.2-fold; SEC61B, 5.5-fold; LMNA, 22.6-fold) (Supplementary Fig. 7). Collectively, our results reveal that Narta can be robustly applied for endogenous gene activation.

### Determinants of Narta efficacy

Next, we sought to investigate the factors that are critical for the gene activation mediated by nascent RNA-guided activators. Because TFs would be recruited to nascent RNAs through MS2-MCP interactions in the Narta system, we tested the effect of MS2 copy number on Narta activation. Toward this end, we constructed four versions of BFP, which contain 0, 6, 12, or 24 copies of MS2 in their introns, for endogenous LMNA and H2B tagging (Supplementary Fig. 8a). By monitoring protein expression, we found that gene activation was strictly limited to the appearance of MS2 sequence in the intron. 12-copy MS2 is likely the best design for Narta activation because it was more efficient than 6-copy and was comparable to 24-copy (Supplementary Fig. 8b, c). Additionally, the DNA size of 12-copy design is smaller than that of 24-copy, possibly making it more suitable for molecular cloning of the donor plasmid. Additionally, the smaller size of NarTag may facilitate higher successful rates of CRISPR-mediated knockin (Supplementary Fig. 8d).

We then asked whether the MS2 sequence could be placed in UTR regions (5′ or 3′) instead of introns (Fig. 2a). By direct comparing the effect of MS2 positions on the activation of four endogenous genes (HSPB1, HSPA1A, LMNA and SEC61B), we observed that intron MS2 is essential for the genes to be highly activated. For example, 3′ UTR-MS2 activated HSPB1 protein expression by 2.5-fold, while intron MS2 could upregulate HSBP1 by 10.1-fold (Fig. 2b, c). By quantifying RNA abundance, qRT-PCR assays also confirmed the different levels of gene activation via intronic or UTR MS2 (Supplementary Fig. 9). These findings suggest that stdMCP-PH activator binding to intron regions is critical for effective Narta activation.

Since the principle of Narta is to concentrate high local densities of TFs by the introns of nascent RNAs, we examined whether PP7/PCP, as another RNA tagging system[49], could function to achieve Narta. We generated a new BFP tag which harbors an intron containing 12 copies of PP7 (termed BFP[PP7-NarTag]). To test whether the artificial intron affects the protein expression of target genes, we inserted BFP, BFP[TriTag (MS2-NarTag)] and BFP[PP7-NarTag] into the N- or C- terminal of target genes (HPDL, HSPA1B, HSPB1, ACTB, SEC61B, and LMNA) by CRISPR knockin (Fig. 2d). We found that PP7-intron in BFP[PP7-NarTag] did not significantly affect the protein expression of all target genes, while MS2-intron in BFP[MS2-NarTag] affected two of them, including HSPA1B (reduced by 44.6%) and HPDL (reduced by 35.3%) (Supplementary Fig. 10). Importantly, we found that Narta activation through PP7/PCP system is as efficient as MS2/MCP (Fig. 2e).

Catalytically-dead Cas13 (dCas13) that retains RNA binding affinity has been engineered for labeling endogenous RNAs in living cells[50,51]. The Cas13b ortholog from *Prevotella* sp. *P5-125* (PspCas13b) was identified as an efficient RNA targeting protein[52]. We fused dCas13b with VPR activator to generate dCas13-VPR. To test whether dCas13-VPR could trigger gene activation, we designed and generated one guide RNA (gRNA) complementary to a repetitive sequence (12 copies of Target Sequence 1, termed 12xTS1) in the intron of BFP[TriTag], thus enabling the binding of multiple dCas13-VPR molecules (Fig. 2f). A non-targeting gRNA was used as a negative control. Our results indicated that dCas13-VPR led to lower levels of gene activation than stdMCP-PH (Fig. 2g). It was reported that dCas13/gRNA exhibited a lower RNA binding affinity than MCP (Kd: MCP < 1 nM; Cas13 ≈ 10 nM)[53,54]. Therefore, the binding affinity between TFs and nascent RNAs may be crucial for Narta. These results demonstrated that Narta activation guided by native introns needs to be further optimized.

### Gene activation mediated by Narta is reversible, tunable, and specific

Next, we sought to test the reversibility and tunability of Narta-mediated gene activation. Using Dox-inducible temporal stdMCP-PH expression system, we monitored the expression of stdMCP-PH and the target gene simultaneously. We found that adding Dox for 2 days could induce high expression of stdMCP-PH, which was then completely degraded after removing Dox for ~5 days (Fig. 3a). By monitoring the expression of miniCMV-driven BFP[TriTag] transgene and two endogenous reporters

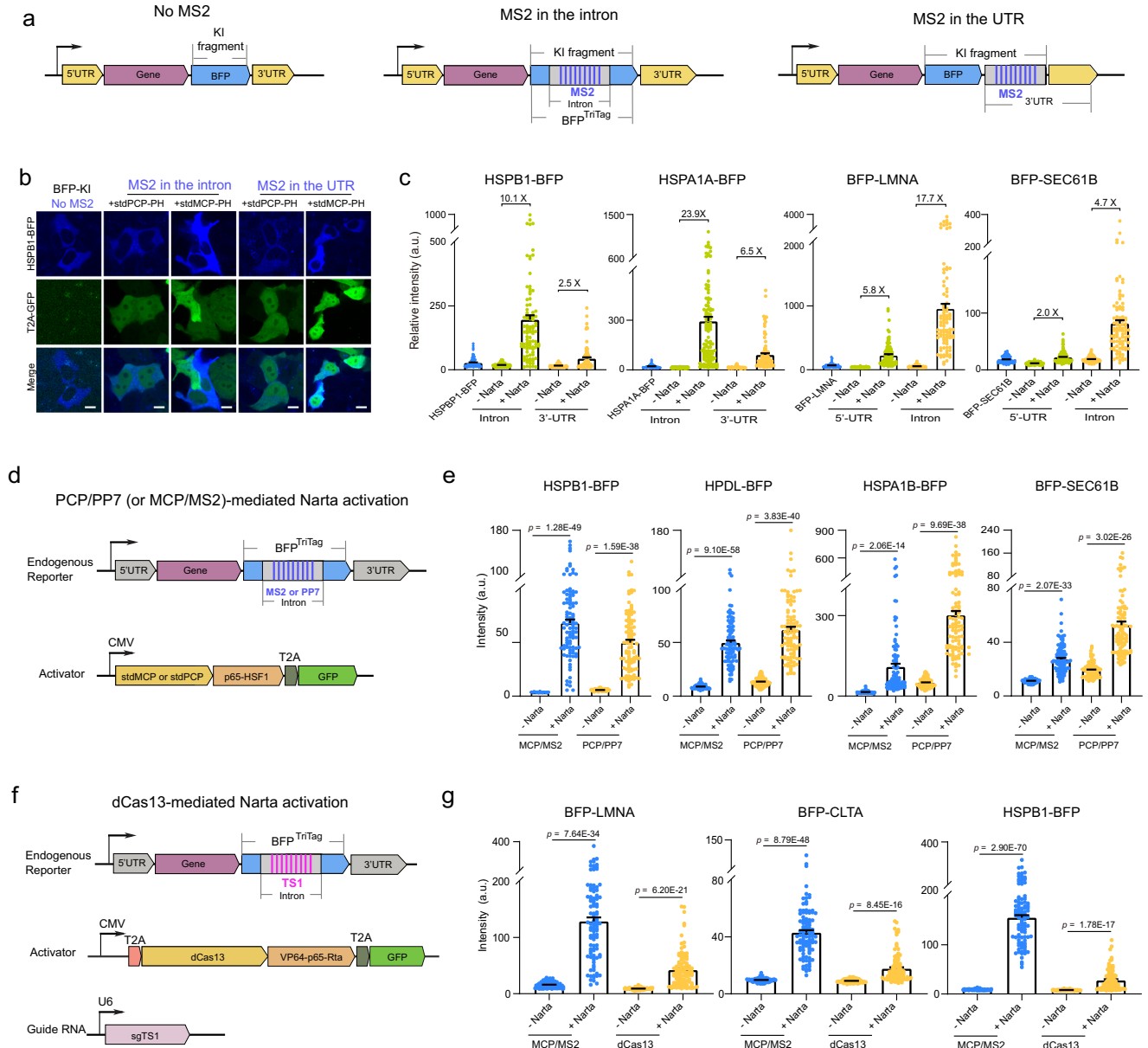

**Fig. 2 | Factors that contribute to Narta efficiency. a** Schematic diagram of different knockin fragments and two different MS2 positions (in the intron or UTR region). **b** Representative images to show HSPB1 expression under various conditions, including different BFP tagging strategies illustrated in **a**. Scale bar, 10 μm. **c** Quantifications of protein expression based on fluorescent imaging under different conditions as shown in **a**. Each dot represents a single cell. *n* = 90 cells. **d** Schematic of the components to mediate Narta activation by PCP/PP7 or MCP/ MS2. **e** Quantitative analysis of the protein expression level of endogenous reporters based on fluorescent imaging. Each dot represents a cell. *n* = 100 cells. **f** Schematic construction designs for testing dCas13-mediated Narta activation. **g** Performance of dCas13-mediated Narta activation as measured by quantitative imaging of endogenous reporters. Each dot represents a single cell. *n* = 100 cells. Data in **c**, **e**, **g** are shown as mean ± s.e.m. *P* values were calculated by two-tailed Student's *t*-test. Source data are provided as a Source Data file.

(H2B-BFP^TriTag & BFP^TriTag-LMNA), we found that Narta's target gene could achieve the maximum expression 2–3 days after the peak expression of stdMCP-PH. Furthermore, the activated expression could be reversed to basal level after removing Dox for about 6 days when stdMCP-PH was nearly depleted in the cells. These findings indicate that Narta-mediated gene activation is fully reversible.

In addition, we tested whether gene activation induced by Narta is tunable by titrating stdMCP-PH plasmid amounts for cell transfection. By examining three endogenous reporters, we found that activation potency is dependent on stdMCP-PH dosage (Fig. 3b–d). However, the optimal dosage is gene-dependent. For example, BFP^TriTag-LMNA achieved the most efficient up-regulation in low dosage, while high dosage reduced activation levels (Fig. 3b). H2B-BFP^TriTag was less sensitive to the dosage of stdMCP-PH (Fig. 3c). In contrast, BAG3

activation highly depended on stdMCP-PH dosage (Fig. 3d). These results suggest that Narta-mediated gene activation is tunable.

An important concern for the use of Narta is its targeting specificity. To evaluate Narta specificity, we performed RNA-seq analysis in HeLa cells. For these experiments, we chose LMNA and HSPB8 as our target genes. We found that the correlation in gene expression between Narta and control samples (transfected with stdPCP-PH) was very similar (*R* > 0.99 in each case), indicating that gene expression is not broadly affected by Narta. LMNA and HSPB8 were the most highly up-regulated genes (4.7-fold and 5.7-fold) in each group, respectively (Fig. 3e). Genes were kept in the analysis if they were sufficiently expressed (CPM > 1) in the control group. We did not observe significant activation of any off-target gene compared to control samples for HSPB8 at the transcriptome-wide level. For the LMNA gene, we

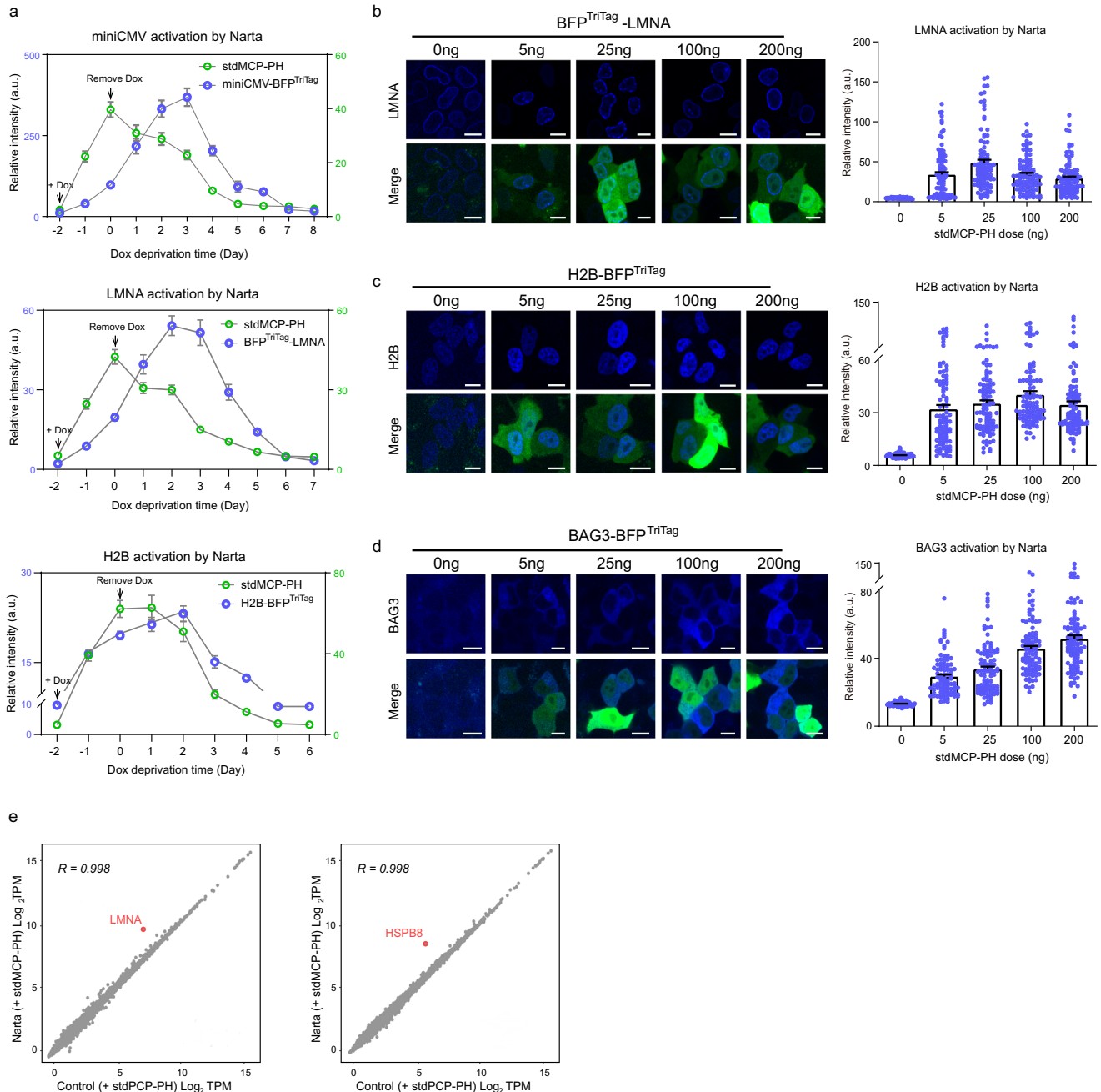

**Fig. 3 | Narta-mediated gene activation is reversible, tunable and specific.**
**a** Green curves showing the expression level of stdMCP-PH-T2A-GFP controlled by the doxycycline-inducible system at indicated time points. The time to add or remove Dox was indicated by arrows (Day −2 and Day 0, respectively). The blue curves illustrate the protein expression level of exogenous (miniCMV) or endogenous (LMNA and H2B) reporters that were related to the expression of stdMCP-PH-T2A-GFP. The protein expression level was quantified based on fluorescent imaging. Each circle represents the mean intensity of the fluorescent reporter of 100 cells at each time point. Error bar denotes mean ± s.e.m. **b–d** Representative images (left) and quantifications (right) to show the does-dependent effect of

stdMCP-PH on Narta activation of three endogenous reporters. Each dot represents a cell. $n = 100$ cells. Scale bar, 10 μm. Data are shown as mean ± s.e.m. **e** Plots to show gene expression levels (log$_2$TPM) in reporter cells (Left: BFP$^{TriTag}$-LMNA; right: HSPB8-BFP$^{TriTag}$) transfected with stdPCP-PH ($x$ axis) versus expression in cells transfected with stdMCP-PH ($y$ axis). Mean values of TMP in two replicates were calculated and log2 transformed to show the expression level of each gene. $R$ indicates Pearson's correlation coefficient, calculated for long-transformed values on all genes except the target gene. Target gene mRNAs are marked in red dots, which are the most significant differentially expressed genes ($t$-test $q$ value < 0.05 with FDR correction). Source data are provided as a Source Data file.

found only one neighboring gene, MEX3A, was differently expressed (2.9-fold, FDR $q$ value < 0.05). Notably, LMNA and MEX3A are transcribed in opposite directions, and their transcription start sites are only 107 bp apart. Therefore, they may share the promoter and transcriptional regulatory complexes. These results suggest that Narta-mediated gene activation is specific with minimal off-target activity.

## Combinatorial use of Narta and CRISPRa can amplify gene activation
CRISPRa activates gene expression by directing aTFs to the promoters or enhancers through limited DNA binding sites[55,56]. Narta, on the other hand, induces high levels of transcription by concentrating abundant aTFs in the transcription sites. Thus, we envisioned that Narta might work more efficiently than CRISPRa if the target gene is not completely

silenced. Because CRISPRa efficiency is heavily dependent on the selection of sgRNAs, we used sgRNA designing tool[57] to select eight or nine sgRNAs for each target gene (BAG3, HPDL, LMNA, TOMM70A, H2B and HSPB1). We picked the three most efficient sgRNAs by comparing the activation efficiency of individual sgRNAs via measuring protein expression (Supplementary Figs. 11 and 12a). As expected, the combinatorial use of the three sgRNAs significantly enhanced the efficiency, revealing the synergistic effects between multiple sgRNAs (Supplementary Fig. 11).

CRISPRa efficiency can be optimized by recruiting multiple activators. Based on this principle, Cas9-SunTag-10XPH (SPH) and dCas9-VPR were developed and represent potent dCas9 activator systems[41,58]. Therefore, we compared the activation efficiencies of SPH, VPR and Narta (Fig. 4a). For each target, CRISPRa (SPH or VPR) was performed with three sgRNAs delivered in concert, but their activation efficiency was much lower than Narta except BAG3 and TOMM70A. We also compared CRISPRa and Narta by examining the activation of miniCMV which presents a weak promoter. Narta activated miniCMV more strongly than CRISPRa (3.8-fold vs. 12.7-fold). Important to note, we found that combinatorial use of Narta (stdMCP-PH) and CRISPRa (dCas9-VPR) significantly enhanced the activation of miniCMV, BAG3, LMNA and TOMM70A (Fig. 4b; Supplementary Fig. 12b). Some results have been confirmed by FACS analysis (Fig. 4c). Taken together, our results demonstrate that combinatorial use of Narta and CRISPRa might be able to achieve the maximum activation of a target gene.

### Narta improves fluorescence-based cell isolation and super-resolution imaging of endogenous reporters

Of note, we used fluorescence-activated cell sorting (FACS) to isolate positive cells with TriTag knockin. Based on FACS analysis, TriTag knockin efficiency in HeLa cells is typically between 0% and 0.5%. Factors affecting the fluorescence detection rate include endogenous protein abundance and gene editing efficiency[59]. Therefore, we hypothesized that temporary expression stdMCP-PH could increase the protein expression level and thus enhance the detection rate of FACS. As expected, we found that Narta activation could greatly enhance FACS sorting efficiencies (e.g., 0.2% vs 4.7% for HSPA1A; 0% vs 2.3% for HSPA1B; 0.3% vs 7.8% for BAG3; 0.2% vs 4.3% for HPDL) (Fig. 5a). Therefore, Narta can be applied to isolate endogenous reporter cells with higher efficiency.

It is challenging to detect a target protein at a low expression level using fluorescent imaging, especially live or super-resolution imaging, which typically needs to acquire multiple frames and is dependent on higher light dose[60,61]. Therefore, photobleaching and phototoxicity have been the major issues. We therefore sought to test whether Narta could improve super-resolution imaging of fluorescent protein-fused endogenous proteins. The endoplasmic reticulum (ER) translocon complex protein SEC61B is traditionally used as an ER marker[62,63]. By applying live-cell Hessian structured illumination microscopy (Hessian-SIM)[64], the location of endogenous BFP[TriTag]-SEC61B to a membrane network of tubules and sheets was clearly visible by SIM imaging only upon Narta activation. However, the endogenous signal without activation was too weak to show structural features (Fig. 5b). Additionally, we imaged endogenous BFP[TriTag]-CLTA (clathrin light chain A). CLTA is the main structural component of coated pits and vesicles (about 150–200 nm in size) which function in the receptor-mediated endocytosis[65]. Only with Narta activation, Hessian-SIM could efficiently detect BFP[TriTag]-CLTA signal and resolve the nanomorphology of clathrin-coated structures (CCSs) (Fig. 5c, d). The size of CCSs measured by Hessian-SIM imaging was similar with a previous report[66] (Fig. 5e). Our results suggest that Narta would be an effective remedy when the fluorescent signal of endogenous fluorescent reporters is too weak for microscopy detection.

### Narta concentrates transcriptional coactivators at target sites

Having established Narta as a gene activation tool, we sought to address how Narta activates the transcription of target genes. The transactivation domains (TADs) of eukaryotic TFs are thought to interact with a set of coactivator complexes, which include Mediator and p300[67–69]. The lysine acetyltransferace activity of p300/CBP mediates BRD4 recruitment to their acetylated sites to promote transcription[26]. Therefore, to examine whether Narta induced local high-concentration interaction hubs at the target genomic loci using the Dox-inducible Narta system, which stably expressed miniCMV-BFP[TriTag], stdMCP-tdTomato and Dox-inducible stdMCP-PH in HeLa cells. We evaluated the subcelluar localization of MED1 by fixed cell immunofluorescence (IF). MED1 imaging indicated that the percentage of visible transcription sites (labeled by stdMCP-tdTomato) enriched for MED1 signals was increased from 6.33% to 93.67% upon Narta activation. In the meanwhile, the intensity of the MED1 signal was increased by 3.31 fold (Fig. 6a, b).

To investigate whether p300 and BRD4 are involved in Narta activation, we tagged endogenous p300 and BRD4 with HaloTag in the Narta-inducible cell line by CRISPR knockin. As expected, stdMCP-tdTomato spots, representing newly transcribed RNAs, appeared in response to the addition of Dox, accompanied by the enrichment of HaloTag-p300 and HaloTag-BRD4 signals in the visible transcription sites (Fig. 6c–f). Moreover, we found a tight positive correlation between nascent RNA production and p300/BRD4 signal (Pearson correlation coefficient, $r = 0.58$ for p300 and $r = 0.24$ for BRD4), suggesting that the recruitment of coactivators to the transcription site is related to the abundance of nascent RNAs (Fig. 6g, h). Next, to address whether p300 and BRD4 play critical roles in Narta activation, we monitored nascent RNA production following A-485 or JQ1 incubation. A-485 inhibits p300-mediated histone acetylation, while JQ-1 is a BRD4 inhibitor[70,71]. Following A-485 or JQ1 treatment, BRD4 enrichment in transcription sites was dramatically reduced (Fig. 6i). At the same time, we observed that transcriptional activation by Narta was significantly repressed with the addition of A-485 or JQ1, suggesting that Narta activation of miniCMV is dependent on the role of BRD4 and p300's histone acetylation activity.

We then analyzed whether stdMCP fused to other aTFs could still regulate transcription. To do so, we fused stdMCP with a series of aTFs, which harbors single (VP64, p65, Rta, HSF1), bipartite (p65-HSF1) or tripartite (VP64-p65-Rta) ADs that had been ever tested for CRISPRa[9,10,15,58]. Their potency to activate transcription was assessed by quantitative analysis of BFP expression based on imaging. stdMCP-VP64, stdMCP-p65 and stdMCP-Rta showed meaningful reporter induction. Consistent with CRISPRa, tandem and bipartite fusions, including stdMCP-p65-HSF1 and stdMCP-VP64-p65-Rta (hereon referred to as stdMCP-PH and stdMCP-VPR) had further improved Narta activation efficiency (Fig. 6j). Of note, there were no changes to BFP expression in cells treated with stdMCP-p300, demonstrating the critical role of TFs in p300's contribution to gene activation.

## Discussion

In summary, our results provide proof of principle that artificial TFs can be highly concentrated in the transcription sites via their binding to introns in the newly transcribed RNAs of the target gene, which then subsequently induce high levels of transcription. That is, nascent RNA can function as a regulator of its own expression. The major advantage of this system is its superior transcriptional activation capacity. Of most genes we tested, Narta showed a much higher activation capacity than CRISPRa. This might be due to the restricted TF binding sites in the promoter region using CRISPRa, while the number of TF binding sites on nascent RNAs of a transcribing gene could be increased by an order of magnitude. Notably, our results demonstrate that Narta and CRISPRa can be combined to further enhance activation efficacy, possibly because they recruit aTFs in different ways. Therefore, the

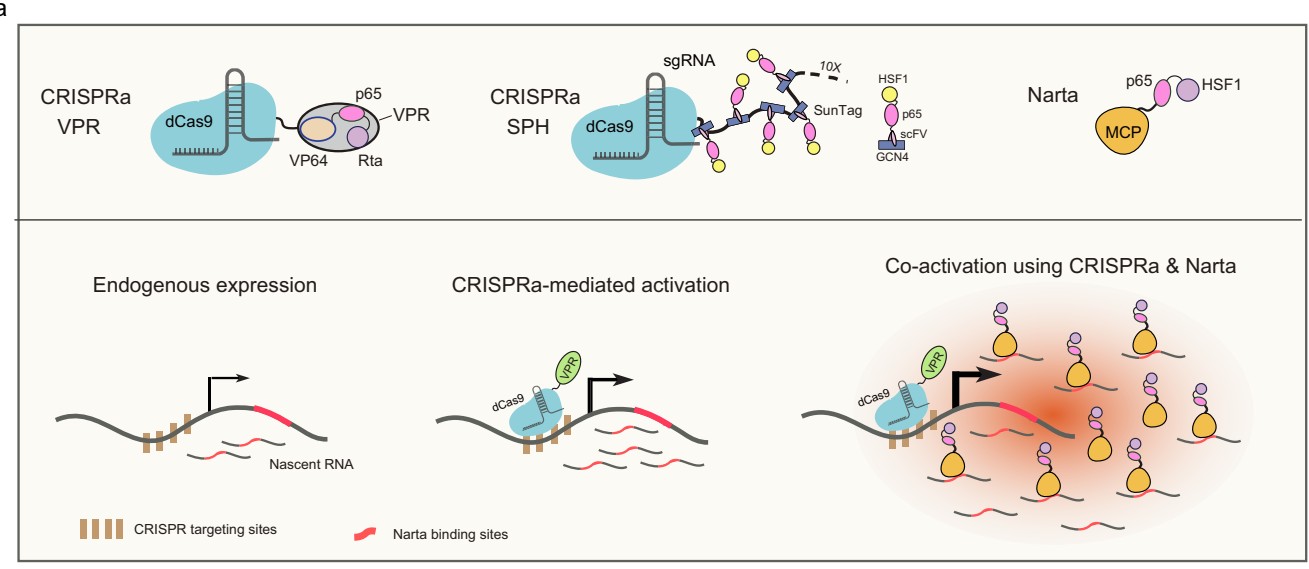

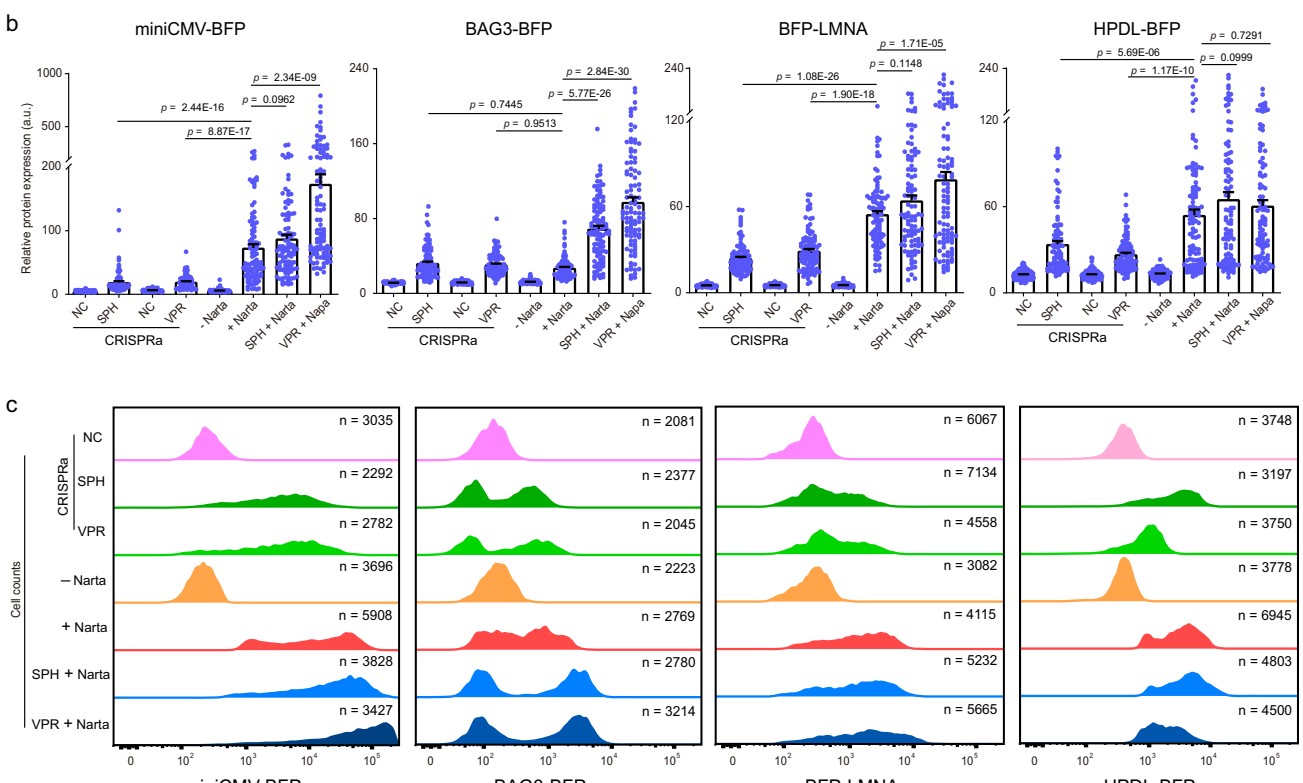

**Fig. 4 | Comparison and combinatory use of CRISPRa and Narta. a** Schematic of different activators and the potential enhancing effect on gene activation between CRISPRa and Narta. **b** Quantifications of protein expression level based on fluorescent imaging under various conditions, including negative controls and gene activation mediated by CRISPRa, Narta or combinatorial use of both. sgGal4 was used as the negative control of CRISPRa systems, while sdPCP-PH (T2A-GFP) was transfected serving as the negative control of Narta activation. Each dot represents

a single cell. *n* = 100 cells. Data is shown as mean ± s.e.m. *P* values were determined by One-way ANOVA with Tukey's post hoc. **c** Flow cytometry quantification of target protein expression levels under the same conditions as in **b**. Transfection-positive cells were gated based on internal reporters (GFP or HaloTag). The distribution of these cells was plotted based on the fluorescence intensity of BFP fused to the target gene (*y axis*). *x axis* is the number of cell counts. Source data are provided as a Source Data file.

combinatorial use of Narta and CRISPRa has great potential benefits for bioengineering and synthetic biology.

Recent studies have presented evidence to support the idea that nascent RNA has an active role in regulating transcription[72,73]. The transcription factor YY1 interacts with nascent eRNA and nascent pre-mRNA discovered by CLIP. It was suggested that the function of these nascent RNAs is to "trap" YY1 surrounding DNA, leading to increased

local concentration of YY1 and facilitating its loading onto neighboring DNA[34]. Similarly, the action of Narta likely creates a positive-feedback loop in which aTFs induce the transcription of nascent RNAs, which then further retain more aTFs locally. Our results reveal that tethering activators to the intron of nascent RNAs is much more efficient than its UTR region, suggesting that the activator-bound intronic RNAs (probably spliced introns) may play a critical role in Narta. We

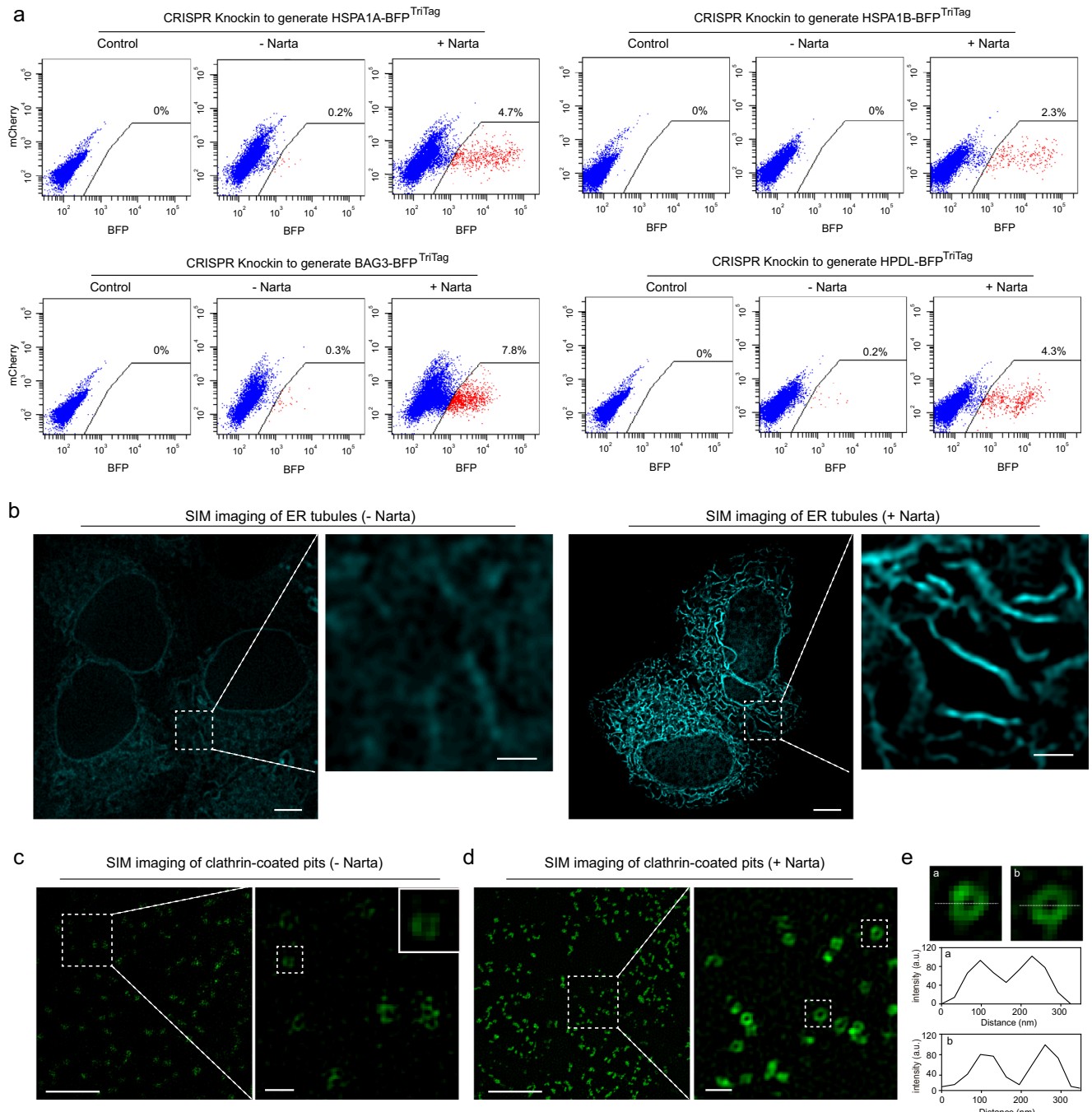

**Fig. 5 | Applications of Narta in fluorescence-based cell sorting and super-resolution imaging. a** HeLa cells were co-transfected with TriTag^BFP knock-in plasmids (including donor, Cas9 and sgRNA) and stdMCP-PH (+Narta) or stdPCP-PH (− Narta). The control group was transfected with sgGal4 instead of sgRNAs targeting genes of interest. Successfully edited cells were selected by FACS and plotted based on the intensity of BFP fluorescence. **b** SIM imaging of ER tubules labeled by SEC61B-GFP^TriTag with or without Narta activation. Scale bar, 5 μm. The boxed regions are further displayed in zoomed-in views with 1 μm scale bar.

**c**, **d** SIM imaging of clathrin-coated pits labeled by CLTA-GFP^TriTag with or without Narta activation. Scale bar of the large field images is 5 μm. The region in the white box is further displayed in a zoomed-in view with 500 nm scale bar. Magnified view of a representative donut-shaped clathrin-coated vesicle from **c** is shown.
**e** Magnified view of two representative donut-shaped clathrin-coated vesicles from **d**, left. Line scan of the relative fluorescence of CLTA-GFP^TriTag is generated to show the size of clathrin-coated vesicles.

speculate that spliced intronic RNA-TFs may be retained in the transcription center and function to mediate transcriptional activation. The underlying mechanisms of Narta activation need to be further investigated. The advantage of nascent RNA having a regulatory role in its own transcription is that it provides precise regulation per se and positive feedback to regulate transcription. Altogether, accumulating evidence suggests that protein-coding mRNA can fulfill additional noncoding functions[74–76]. Our work suggests that the awareness of RNA

bifunctionality is not only of conceptual importance, but will be increasingly useful when nascent RNA acts as a general binding platform to recruit transcription regulators.

Previous studies have suggested that TF concentration regulates transcriptional bursting kinetics[26,43,77]. Moreover, strong enhancers drive bursts at a higher frequency than weak enhancers, while SEs exhibit similar bursting patterns to strong enhancers, showing relatively longer burst duration and higher burst

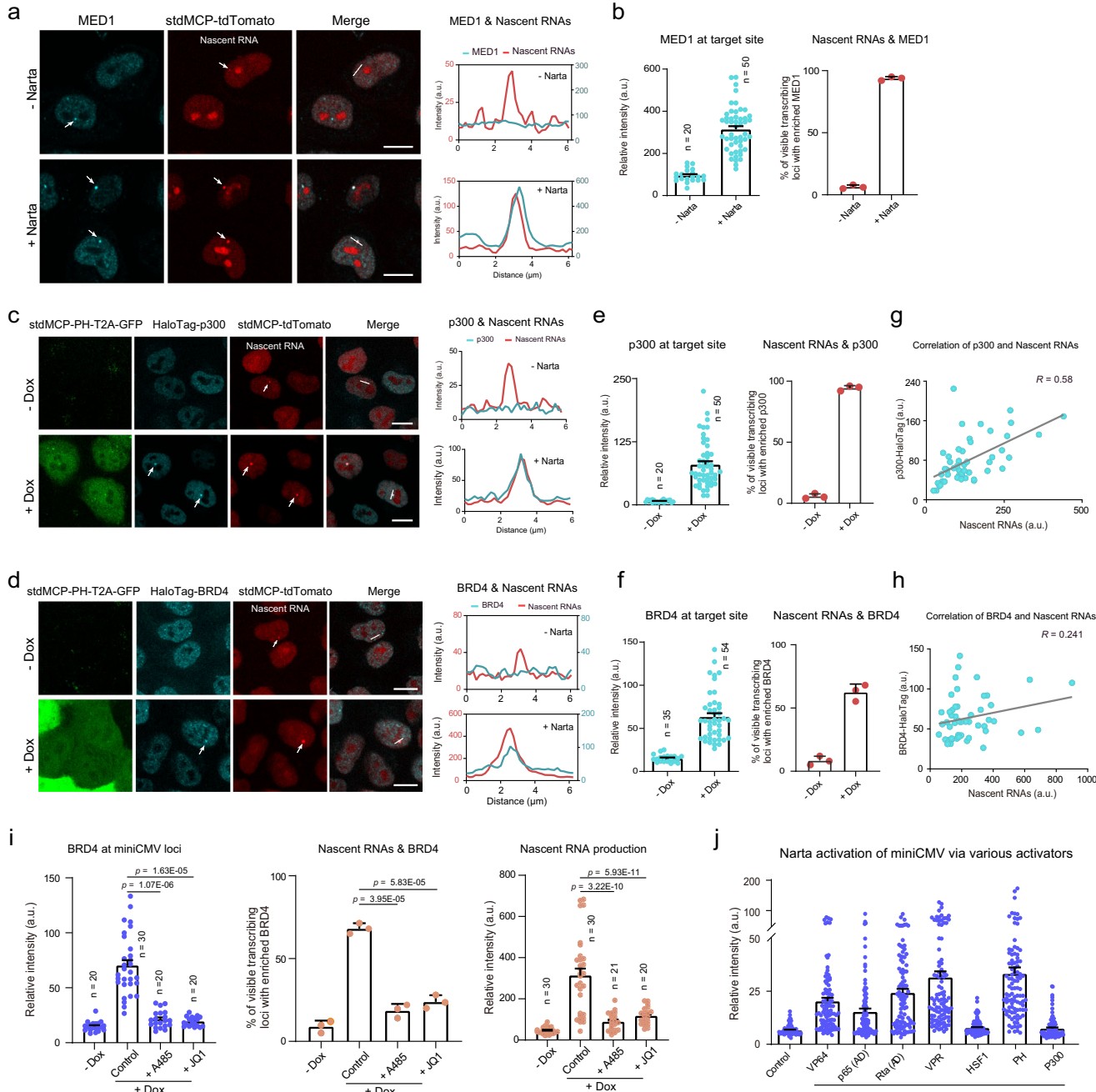

**Fig. 6 | Narta can induce high-density assembly of coactivators at target sites.**
**a**, **c**, **d** Left, representative images to show the co-localization between coactivators (MED1, p300, or BRD4) and nascent RNAs (labeled by stdMCP-tdTomato) produced by miniCMV. Right, line scan of the relative fluorescence intensity indicated by the dotted lines in the left panel. In **a** stdMCP-PH was transfected to induce Narta activation, while stdPCP-PH tranfection was used as the negative control. In **d**, **f** the expression of stdMCP-PH was induced by the addition of Dox. MED1 was detected by antibody, while p300 and BRD4 were endogenously tagged with HaloTag. Scale bar, 10 μm. **b**, **e**, **f**, Left, total intensity of activator reporters (MED1-Alex647, HaloTag-p300 and HaloTag-BRD4, respectively) enriched at visible miniCMV transcribing loci. Right, quantifications showing the percentage of visible stdMCP-stdTomato spots (representing active mimiCMV loci) which were co-localized with enriched signal of coactivators (MED1, p300 or BRD4). n = 3 biological replicates.

**g**, **h** Scatter plots of nascent RNA level (x-axis) and the enriched signal of co-activator (p300 or BRD4, y-axis). Gray line denotes the linear fit. R represents the correlation coefficient. Each dot represents a single cell. n = 50 cells.
**i** Quantifications of BRD4-HaloTag signal (right), nascent RNA production (total intensity of individual stdMCP-tdTomato foci, left), and the co-localization between nascent RNA and BRD4 (n = 3 independent experiments by examining 30 cells in each repeat) at active miniCMV loci under different conditions. Dox was added to induce Narta activation for 12 h. Together with Dox, DMSO, A-485 or JQ1 was added into the medium. P-value was analyzed by One-way ANOVA with Tukey's post hoc.
**j** Quantifications of miniCMV-BFPTriTag expression levels in cells transfected with stdPCP-PH (negative control), stdMCP-p300 or different stdMCP-TFs (n = 100 cells). All histograms in Fig. 6 are displayed as mean ± s.e.m. Source data are provided as a Source Data file.

amplitude[23,78]. Following Narta activation, the transcriptional bursting dynamics of both miniCMV and H2B promoters were altered, displaying similar burst characteristics to those of the strong enhancers. Although Narta is an artificial manipulation, its regulatory effect on genes suggests that this system modulates transcriptional bursting kinetics in a similar manner to the behavior of endogenous transcription factors.

Labeling endogenous proteins with fluorescent tags is valuable for studying protein function in a native cellular background[79,80]. CRISPR-Cas9 editing is a powerful tool to introduce a fluorescent tag into a target gene by homologous recombination[59]. Fluorescent protein-tagged cells can be isolated by flow cytometry (FACS), but it is highly dependent on protein abundance. Furthermore, the knockin efficiency could be very low. Our results reveal that the fusion of NarTag (Tag for Narta activation, which is a fluorescent protein tag harboring an artificial intron incorporated with MS2 or PP7 sequence) did not affect the protein expression of most target genes. Therefore, using NarTag instead of commonly used fluorescent tags to generate endogenous reporters is a promising alternative strategy. Transient activation by Narta will greatly increase the FACS efficiency. Many studies have indicated that protein abundance is a major limiting factor for microscopy detection[59,81]. Although most of our targets are highly-expressed genes in the human genome (FPKM values range from 6 to 950 with a median value of 41.5 based on our RNA-Seq data), endogenous fluorescent tagging of these genes still suffers from poor signal and photobleaching. Our results indicate that the use of Narta can manipulate protein abundance as required for super-resolution imaging and 3D time-lapse fluorescence microscopy (4D imaging). Notably, overexpressed proteins may lead to artifacts, including mislocalizations and protein aggregation[82]. Because Narta is a tunable activation tool, different levels of gene activation can be considered when implementing Narta.

On the other hand, the major limitation of current Narta is the requirement of genetic modification of target genes. Therefore, it is not applicable for the genome-wide genetic screen as CRISPRa. An RNA-binding protein that can direct activators to endogenous introns would be an ideal system to achieve Narta. dCas13-VPR can activate gene expression by using a highly efficient crRNA which has 12 binding sites on the artificial intron of target genes. However, the activation potency is much lower than stdMCP-PH, which might be due to its weaker RNA-binding affinity than MCP[83]. Therefore, further optimization of dCas13-activator system may solve the issue of genome engineering. Moreover, Narta targets may be limited to transcriptionally-active genes. However, this obstacle can be addressed by implementing CRISPRa and Narta simultaneously. Finally, we found that Narta could mediate gene activation with high specificity. However, co-regulated genes that colocalize in the same transcriptional factory[84] might be co-activated by Narta simultaneously. This should be taken into consideration in implementing Narta. In conclusion, Narta represents a complementary strategy to induce overexpression of target genes in addition to cDNA overexpression and CRISPRa (Supplementary Fig. 13). Notably, tagging endogenous genes with NarTag enables dynamic imaging of gene expression at the levels of RNA and protein and manipulating gene expression with tunable capability in a single cell.

## Methods

### Cell culture
HeLa cells and HEK293T cells were maintained in Dulbecco's Modified Eagle's Medium (DMEM) with high glucose (Gibco) in the presence of 10% FBS (Hyclone) and 1% penicillin/streptomycin (Gibco). CHO-K1 cells were cultured in Hams F-12K Nutrient Mixture (Kaighn's) supplemented with 10% FBS and 1% penicillin/streptomycin. Cells were maintained in a humidified incubator set at 37 °C and 5% $CO_2$. All cell lines were regularly tested for mycoplasma.

## Plasmid construction
The assembly of TriTag has been described in our previous study[39]. Other plasmids were constructed specifically for this study as following:

**Construction of Narta activators.** To build stdMCP-PH (p65-HSF1)-T2A-GFP and stdPCP-PH-T2A-GFP, the DNA sequences encode stdMCP, stdPCP or PH-T2A-GFP were amplified from Addgene plasmids #164044[39], #104099[85] and #107311[41], respectively. Fragments of stdMCP (or stdPCP) and PH-T2A-GFP were assembled into a lentiviral vector harboring CMV promoter using NEBuilder HiFi DNA Assembly Cloning Kit (New England Biolabs). In addition to GFP, BFP, mCherry and HaloTag were also constructed to represent the expression of activators. Moreover, stdMCP-VP64-T2A-mCherry, stdMCP-p65-T2A-mCherry, stdMCP-Rta-T2A-mCherry, stdMCP-VPR-T2A-mCherry, stdMCP-HSF1-T2A-mCherry, and stdMCP-p300-T2A-mCherry were constructed using the same strategy. The cDNA of p300 was amplified from the genomic DNA of HEK293T cells. The fragments of VP64, p65 and Rta were amplified using VPR (#63798)[58] as the PCR template. To generate Dox-inducible Narta, the fragment of stdMCP-PH-T2A-GFP was cloned into a lentiviral vector harboring an inducible TRE3G promoter (Tet-on 3G inducible expression system, Clontech).

**Construction of dCas-activators.** To build dCas9-VPR, dCas9-SPH (consist of dCas9-10XGCN4 and scFV-PH) and dCas13b-VPR, the DNA sequences that encode VPR, dCas13b, scFV-PH, 10XGCN4 were amplified from Addgene plasmids #63798[58], #103866[52], #107311[41] and #107307[41], respectively. Corresponding fragments were assembled into a lentiviral vector harboring CMV promoter by NEBuilder HiFi DNA Assembly Cloning Kit (New England Biolabs).

**Construction of exogenous reporters to test Narta activation.** To test Narta activation efficiency, the exogenous reporter BFP[TriTag] was driven by different promoters. To do this, BFP[TriTag] was cloned into a vector harboring miniCMV, SFFV, CMV, CAG, or EF1α promoter using T4 DNA ligase (New England Biolabs).

**Construction of donor plasmids for CRISPR knockin.** To label an endogenous gene via a TriTag or a conventional fluorescent protein, CRISPR-mediated homology directly repair (HDR) was implemented. All the donor plasmids were constructed using a same strategy. Using the H2B gene as an example, fragments of the left and right homology arm (HA) of H2B and the TriTag (BFP) were assembled into a vector to generate 5'HA-TriTag-3'HA using NEBuilder HiFi DNA Assembly Cloning Kit. Notably, to increase knock-in efficiency, a Cas9 cleavage site (GGAGCTTACTGAGACTCTTC<u>GGG</u>, Targeting sequence 2 including PAM termed TS2) was included in the forward primer of left HA and the reverse primer of right HA to generate a double-cut donor plasmid[39,86].

**Construction of sgRNA and crRNA plasmids.** To generate sgRNA expression plasmids, we used our previous sgRNA vector (Addgene #164043) which harbors mouse U6 promoter, spacer sequence, and the optimized sgRNA scaffold (E + F)[87]. The spacer sequence can be easily replaced to recognize other targets by the PCR-based QuikChange cloning method. sgRNAs used for Cas9 editing and CRISPRa were designed using the sgRNA designing tool (http://chopchop.cbu.uib.no/). The crRNA for Cas13 targeting were built using a cloning strategy similar to that of sgRNA. The sgRNA and crRNA used in this study are listed in Supplementary Tables S1 and S2.

## Lentivirus production and generation of Dox-inducible Narta cell lines
To construct the Dox-inducible Narta cell line, the lentivirus production assay was performed for TRE3G- stMCP-PH-T2A-GFP and CMV-

Tet-On 3G, respectively. To do this, HEK293T cells were plated on 12-well plates. The next day, 705 ng pCMV-dR8.91 and 87 ng PMD2.G together with 750 ng target plasmid were delivered into HEK293T cells using FuGENE (Promega) following the manufacturer's recommended protocol. Virus was collected 60 h after transfection and then centrifuged at 800 $g$ for 10 min. The supernatant was added directly to target cells or stored at −80 °C. To generate Dox-inducible Narta cells, HeLa cells with or without stable expression of stdMCP-tdTomato were infected with the mixture of TRE3G- stMCP-PH-T2A-GFP and CMV-Tet-On 3G lentiviruses. To improve the infection efficiency, polybrene (5 µg/ml) was added into the medium of target cells during infection. stdMCP-PH-T2A-GFP positive cells were then isolated by FACS selection in the presence of Dox.

## CRISPR-mediated knock-in

To label a specific endogenous gene with a fluorescent tag (e.g., HaloTag, BFP, BFP[TriTag], or GFP[TriTag]), HeLa or 293T cells were seeded into 24-well plates and then transiently transfected with plasmids including 100 ng Cas9, 400 ng donor, 250 ng sgRNA of a target gene and 250 ng sgTS2 expression vectors. 2–3 days later, CRISPR knockin positive cells were selected by FACS analysis. Other CRISPR knockin experiments were performed using the same protocol. To quantify knock-in efficiency in Supplementary Fig. 8, a GFP vector was co-transfected to indicate the successfully transfected cells.

## Gene activation by Narta

To perform Narta activation of exogenous reporters, corresponding cells (HeLa or CHO-K1) were seeded into eight-well chambered coverglass. miniCMV-BFP[TriTag] (200 ng), SFFV-BFP[TriTag] (200 ng), EF1α-BFP[TriTag] (200 ng), CMV-BFP[TriTag] (100 ng), and CAG-BFP[TriTag] (50 ng) were co-transfected with 100 ng stdMCP-p65-T2A-GFP or stdPCP-p65-T2A-GFP into corresponding cell lines. BFP expression was quantified by fluorescent imaging 48 h after transfection. To achieve Narta activation in the Dox-inducible Narta cell line (miniCMV-BFP[TriTag]), doxycycline (1 µg/ml, Sigma-Aldrich) was added 12 h ahead of RNA imaging and 48 h ahead of Protein imaging. For Narta activation of endogenous promoters, TriTag (BFP or GFP) tagging cell lines seeded into eight-well chambered coverglass, 24-well or 6-well plates. 100 ng (8-well), 300 ng (24-well), or 1200 ng (6-well) stdMCP-p65-T2A-GFP and stdPCP-p65-T2A-GFP were transfected into the corresponding cells the next day. To perform dCas13b-mediated Narta activation, 300 ng dCas13b-VPR-T2A-GFP and 600 ng crTS1 were co-transfected to corresponding endogenous reporter cells in eight-well chambered coverglass. Cells in eight-well chambered coverglass were used for confocal microscopy imaging and protein fluorescence analysis 48 h after transfection. To image nascent RNA production, cells in eight-well chambered coverglass were imaged under confocal microscopy 12 h after transfection. Cells in 24-well and 6-well plates were used to perform FACS and RNA extraction, respectively, 48 h after transfection.

## Gene activation by CRISPRa

To perform CRISPRa, cells harboring TriTag (BFP) tagging of a specific endogenous gene were seeded into 8-well chambered coverglass, 24-well, or 48-well plates. Taking 8-well as an example, cells were transiently transfected with 300 ng dCas9-VPR-T2A-GFP (VPR system) or 300 ng dCas9-10XGCN4v4-T2A-GFP and 300 ng scFV-PH (SPH system). In addition, three sgRNAs targeting the same gene were co-transfected at a total amount of 600 ng. To perform co-activation of Narta and CRISPRa in 8-well plates, 100 ng stdMCP-P65-T2A-EGFP were co-transfected with 300 ng dCas9-VPR-T2A-GFP or 300 ng dCas9-10XGCN4v4-T2A-EGFP and 300 ng scFV-PH to corresponding reporter cells. After 48 h transfection, cells were used for confocal microscopy imaging. To perform FACS analysis, cells were seeded in 24-well or 6-well plates 12 h ahead of plasmid transfection and were collected 48 h after transfection.

## Flow cytometry

To isolate CRISPR knock-in positive cells, cells were analyzed and selected by MoFlo Astrios EQ (Beckman). Cells were first gated for the intact cell population based on forward scatter versus side scatter plots and then gated for single cells using forward scatter A versus forward scatter H. Positive cells were sorted out for further validation of CRISPR knock-in. To improve TriTag knock-in efficiency, CRISPR knock-in was performed with additional supplement of 300 ng stdPCP-PH-T2A-EGFP (as negative control) or stdMCP-PH-T2A-EGFP plasmids. Positive cells were gated out using the same protocol to calculate the positive rate of CRISPR knock-in. To compare the activation efficiencies of Narta and CRISPRa, the expression level of target proteins was analyzed by flow cytometry using BD Fortessa instrument (BD Biosciences). Cells that were positive for Narta and CRISPRa expression were gated based on the GFP or HaloTag reporter. Gating strategies were provided in Supplementary Fig. 14.

## Quantitative RT-PCR

To confirm Narta activation at the RNA level, corresponding cells were collected for RNA extraction using FastPure Cell/Tissue Total RNA Isolation Kit (Vazyme) following the manufacturer's instructions. RNA was first converted to cDNA using oligo-dT primers (HiScript II Q RT SuperMix for qPCR, Vazyme). PCR reactions were carried out using ChamQ Universal SYBR qPCR Master mix (Vazyme) and were performed on the QuantStudio 5 Real-Time PCR system (Thermo Fisher). All experiments were repeated three times using samples of three independent batches. The RNA abundance was normalized to an endogenous gene UBC and calculated as delta-delta threshold cycle (△△Ct). Primers used for qRT-PCR are listed in Supplementary Table S3.

## Western blot

HeLa cell lines expressing specific GFP[TriTag] tagged endogenous reporters were plated on 24-well plates, and then transfected with 300 ng of plasmids encoding stdMCP-PH-T2A-GFP or stdMCP-PH-T2A-GFP the next day. Cells were collected 48 h after transfection. Samples were lysed and then loaded into a 10% Bis-Tris protein gel (Thermo Scientific). GFP[TriTag] tagged endogenous proteins were detected with mouse anti-GFP antibody (1:2000, EarthOx, E022280). Actin was detected with β-Actin Rabbit mAb (1:5000, Sangon Biotech, D191047). Blots were imaged by ChemiScope 3300mini.

## Immunostaining

To detect endogenous MED1, cells were fixed with 4% paraformaldehyde, permeabilized with 0.5% NP-40 in phosphate buffered saline (PBS) for 10 min, washed with PBS buffer for 5 min and repeated twice, blocked in 0.2% cold water fish gelatin and 0.5% bovine serum albumin (BSA) for 20 min, incubated with the primary antibody in blocking buffer at room temperature for 4 h, washed three times and then incubated with Alexa647-conjugated secondary antibody at room temperature for 1 h, and finally washed three times again. The primary and secondary antibodies for detecting MED1 are Rabbit anti-MED1 Antibody (1:1000, A300-793A, Bethyl) and Donkey Anti-Rabbit IgG H&L conjugated with Alexa Fluor® 647 (1:1000, ab150075), respectively.

## smFISH to detect Narta activation

smFISH probes complementary to BFP sequence were designed to cover the region of BFP (Cat # CUS288-D1, GD Pinpoease Biotech Co., Ltd.). smFISH was performed using PinpoRNA™ RNA in-situ hybridization kit according to the manufacturer's instructions (Cat #: PIT1000, GD Pinpoease Biotech Co. Ltd.). Briefly, cells were first fixed by 10% NBF and then endogenous peroxidase was inhibited by Pre-A solution at room temperature. Target RNA molecules were exposed by

protease treatment and hybridized with probes for 2 h at 4 °C. The signal was then amplified sequentially by reactions 1, 2, and 3. The HRP molecule was added into reaction 3. Lastly, a tyramide fluorescent substrate (OpalTM520, Akoya Biosciences) was added to the cells and the target RNA was then fluorescently labeled by Tyramide Signal Amplification (TSA) assay. Notably, HaloTag expression was used to indicate the successful transfection of stdMCP-PH (+Narta) or stdPCP-PH (−Narta) plasmids. However, no HaloTag signal was detected after smFISH staining was completed. Thus, we selected the cells with the most significant mRNA staining for further quantification and used the same principle for all samples.

### Zebrafish embryo injection
To test Narta activation in Zebrafish embryos, corresponding plasmids (three mixtures: CMV-mCherry + CMV-GFP + blank vector; CMV-mCherry + CMV-GFP^{Fish_NarTag} + stdPCP-PH; CMV-mCherry + CMV-GFP^{Fish_NarTag} + stdMCP-PH) were prepared for microinjection. The concentration of each plasmid is 100 ng/µl. Zebrafish embryos were collected 15 min after fertilization. 2 nl of plasmids mixed with phenol red (10:1) was injected into the yolk of each embryo at 1-cell stage. Quantitative imaging of GFP and mCherry expression was performed at 80% epiboly to 90% epiboly.

### RNA-seq analysis
To address the specificity of Narta system, BFP^{TriTag}-LMNA and HSPB8-BFP^{TriTag} HeLa cell lines were seeded into 6-well plates, respectively. The next day, 4 µg stdMCP-PH-T2A-GFP or stdPCP-PH-T2A-GFP was transfected into cells. RNA extraction was performed 48 h after transfection. Total RNA samples were therefore collected for RNA-seq analysis. Single-end sequencing (50-bp reads) was performed. The sequencing reads were aligned to the human GRCh38 genome using STAR (https://github.com/alexdobin/STAR). Gene expression counts for each sample were calculated by featureCounts[88]. Genes with low counts, of which CPM (counts per million) ≤1 in both control samples, were filtered out. Normalized gene expression of each gene was obtained by function count from DESeq2 package (https://github.com/mikelove/DESeq2). The edgeR package (https://bioconductor.org/packages/release/bioc/html/edgeR.html) was then used to perform differential expression analyses between control and Narta with default parameters. The differential expression genes (upregulated) were defined by a Benjamini–Hochberg adjusted $p$ value ($t$-test $q$ value < 0.05 with FDR correction) and fold change of >2.0.

### Drug treatment
To confirm the involvement of p300 and BRD4 in Narta activation, we took advantage of A-485 (Selleck Cat# S8740) and JQ1 (SIGMA Cat# SML0974) to inhibit the function of p300 and BRD4, respectively. Cells with doxycycline-inducible Narta system were seeded into eight-well chambered coverglass (Thermo Fisher Scientific) prior to drug treatments. The next day, doxycycline (1 µg/ml, Sigma-Aldrich) and A-485 (10 µM/L) or JQ1 (1 µM/L) were added to corresponding wells. After 12 h, cells were subjected to imaging.

### Confocal microscopy
All confocal images were acquired on an Olympus spinning-disk confocal system SpinSR, equipped with Yokogawa CSU-W1 scanner, an sCMOS camera (Prime 95B), a ×60 NA 1.49 oil Apochromat objective, 405/488/561/640 nm lasers (OBIS), and a PIEZO stage (ASI) with stage incubator (Tokai Hit). To perform live-cell imaging, cells were maintained in a humidified chamber set at 37 °C and 5% $CO_2$. Cells were seeded into 8-well chambered coverglass for confocal imaging. All the images and corresponding data quantifications collected in this study were generated using this protocol.

### HIS-SIM imaging
All super-resolution imaging experiments were performed using commercialized Hessian-SIM, termed HIS-SIM (High Intelligent and Sensitive Microscope) equipped with a 100×/1.5NA oil immersion objective (Olympus). HIS-SIM was provided by Guang zhou Computational Super-resolution Biotech Co., Ltd. Cells were cultured in eight-well chambered coverglass for SIM imaging. The protocol for SIM imaging was performed as described previously[64]. In addition, sparse deconvolution was carried out to improve the resolution and contrast of images[89].

### Statistics and reproducibility
ImageJ software was used to analyze fluorescence imaging data for calculating the mean intensity of fluorescent reporter proteins and the total intensity of fluorescent spots (representing nascent RNA, MED1, p300, or BRD4 signals). Line scan was obtained using the "Analyze/Plot Profile" function (a plugin for ImageJ). The extracted parameters were then analyzed in Excel and plotted using GraphPad Prism. GraphPad Prism (Version 8, GraphPad Software, La Jolla, CA, USA, https://www.graphpad.com) was used to calculate the mean values, the standard error of the mean (SEM) and correlation coefficient ($r$) for the statistical analysis. The statistical significance between two groups was calculated via student $t$ test, and significance among three or more groups was calculated using one-way ANOVA. FACS data were analyzed using FlowJo v10 software (FlowJo LLC). All results were reliably reproduced at least once.

### Reporting summary
Further information on research design is available in the Nature Portfolio Reporting Summary linked to this article.

## Data availability
The data that support this study are available from the corresponding authors upon reasonable request. Raw-data of RNA-Seq has been deposited in the Gene Expression Omnibus (GEO) database under the accession number GSE204666. Key plasmids will be deposited to Addgene. Source data are provided with this paper.

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

## Acknowledgements

We thank Dr. Cong Yi and Dr. Yu Feng for their technical support in testing Narta activation. We thank the members, including Yingying Huang and Yanwei Li, in Core Facilities of Zhejiang University School of Medicine for their technical assistance in FACS sorting experiments and Wenyi Huang from Guanzhou CSR Biotech Co. Ltd for the technical support in using their commercial super-resolution microscope (HIS-SIM). This work was supported by National Key R&D Program of China (2021YFC2700904) and National Natural Science Foundation of China (32171444, 31872819, 31800861 and 31970919).

## Author contributions

W.Z. and B.C. supervised the whole study; Y.L., H.X., P.Q., Y.Y., P.X., W.Z., and B.C. designed the experiments; Y.L., H.X., T.C., Y.F., H.H, W.Q., J.W., Y. Z., and Y.Y. carried out experiments; Y.L., H.X., Y.F., H.H, J.W., and B.C. conducted imaging data analysis; T.C. analyzed RNA-Seq data; Y.L., H.X., and B.C. designed and made the figures; Y.L., W.Z., and B.C. wrote the manuscript from inputs from all authors.

## Competing interests

Y.L., W.Z., and B.C. submitted a patent application (Chinese Patent Application No. 2022111029154) on the design and application of a nascent RNA-guided gene activation method based on the technology developed in this paper. All other authors declare no competing interests.
