## [Peer Review File · Nature Communications]

REVIEWER COMMENTS

Reviewer #1 (Remarks to the Author):

Summary

The authors describe a new eukaryotic gene activation system, called Narta, that relies on recruiting transcription factors to nascent transcribed RNA. The authors use RNA binding proteins (RBPs) fused to transcriptional activation domains. The RBPs are directed to nascent RNAs with either heterologous recruitment sites inserted into the target gene or Cas13-based programmable RNA targeting systems. The authors demonstrate that the system is effective in a wide range of systems and can be enhanced with simultaneous CRISPRa at the corresponding genomic DNA. The strongest activation occurred when heterologous RNA target sites were inserted in an intron, and the effects were more modest when using dCas13 to target unmodified endogenous transcripts. The authors readily acknowledged this issue. Overall, the paper presents a novel new method with potential utility for basic research on transcription and for synthetic gene activation. I recommend publication after the authors address the comments below.

Major Comments

1. I found the direct comparisons between CRISPRa and Narta to be one of the more compelling parts of the paper, and the data in Figure 5 addressed the questions I had from the start of the paper. The authors might consider restructuring their manuscript to put Fig 5 before Fig 4, which in my view was interesting but not nearly as critical for establishing the utility of Narta.

2. The authors provide extensive details about transcriptional dynamics, including burst times and amplitudes. The authors should elaborate on why this information is significant and how it can be interpreted. The authors noted on line 147: "These results suggest that Narta activation induced the production of more H2B transcripts, which was further confirmed by qRT-PCR. Consistent with transcriptional activation, the expression of H2B-BFP was dramatically increased by 7.9-fold upon Narta treatment (Fig. 1e)." Why do we need any information about the bursting patterns if the point is that more transcripts are produced? Is there some unstated significance to the specific changes in burst dynamics? Does it matter that burst duration increased while pauses stayed the same? What would it mean if burst duration was unchanged and pause lengths decreased?

Minor comments

1. Line 27: "Moreover, Narta provides better activation potency of some expressed genes than CRISPRa and, when used in combination with CRISPRa, has a synergistic effect on gene activation." Are the effects truly synergistic - more than would be expected based on the sum of the individual effects? The discussion in the main text starting on line 294 is careful on this point, but the abstract (line 27) and the Fig 5 legend (line 871) imply synergy. Enhanced activation above either individual mechanisms is still valuable even if the effects are not strictly synergistic. Minor wording changes would be sufficient to address this point.

2. On line 44, the authors suggest that new approaches are needed to upregulate highly-expressed genes. Have the authors actually demonstrated that the Narta method is better than other gene activation methods at highly-expressed genes? If so, please elaborate. If not, this point may not be necessary. Improved gene activation methods are useful and activating highly-expressed genes may not be the most compelling application.

3. The authors used MCP-MS2 to recruit activators and for imaging nascent RNA production, which means the activators and the fluorescent probes are competing for the same RNA targets. This strategy seems less than ideal. I certainly would not suggest that the authors repeat all of their experiments with orthogonal strategies, but the authors could acknowledge a potential concern here and discuss strategies to build orthogonal activation and reporting systems.

4. Starting on Line 102 and throughout the manuscript, the authors report values to an absurdly large number of significant figures. What is the significance of a burst duration of precisely 16.71 minutes? Is it meaningfully different than 17 minutes? I recommend that the authors re-evaluate the precision with which they report all of the values in the manuscript.

5. On line 140, the authors write: "Next, to monitor the activation of an endogenous gene, we inserted TriTag into the C-terminus of human H2B..." Opinions could differ here, but once the genome needs to be edited to include a heterologous sequence I would no longer consider the gene to be "endogenous". A tool to activate endogenous genes should ideally target unmodified endogenous genes, which the authors address later in the manuscript. Once you have decided to edit the genome, why not simply introduce a strong inducible promoter?

6. Line 184: "Additionally, the DNA size of 12-copy design is much smaller than that of 24-copy, making it more suitable for CRISPR-mediated knockin." Does the size difference between 12 and 24 MS2 repeats really make a large difference for knockin efficiency? I'd generally agree working with 12x MS2 is easier than 24x MS2, but more because larger numbers of repeats can be harder to clone and genetically manipulate.

7. Line 186: The authors compare Narta recruitment sites in UTRs and introns, and observe that intron recruitment gives stronger activation. Can the authors speculate why introns and UTRs behave so differently? Is this observation telling us anything useful about the underlying mechanism?

8. On line 259, the authors discuss using Narta to overexpress fluorescent reporters for imaging. There are discussions throughout the cell biology literature about whether the behavior and localization of overexpressed proteins accurately reflects their behavior at the endogenous expression level. The authors are encouraged to acknowledge these concerns.

9. Line 355: "Our results reveal that tethering activators to the intron of nascent pre-mRNA is much more efficient than its UTR region, suggesting that the concentration of activator-bound RNAs (probably spliced introns) may play a critical role in Narta." This statement is unclear. Why is the concentration of activator-bound RNAs different if the same number of recruitment sites are displayed in a UTR or an intron?

Reviewer #2 (Remarks to the Author):

In this study, Liang and colleagues report the development of a novel molecular approach that drives gene over-expression in a tunable manner. As opposed to current existing methods that are based on synthetic activators /DNA interactions, this novel method is based on synthetic activators/nascent RNA interactions.

The authors have to be acknowledged for their creativity and their ability to take advantage of the MS2/MCP RNA reporter system, which is a technology traditionally used to study

transcription kinetics in real time.

The work reported here is quite mature and advanced, the science is sound, with overall a lot of carefully designed controlled experiments and appropriate quantitative measurements. The method seems well established, robust, efficient and reproducible. Overall the data presented are fairly convincing and this original method is likely to provide a great adjunct to the current toolbox in the field.

It is great to see that the authors spent a lot of efforts to not only characterize in depth the efficacy of their system but also to benchmark it against other technology (such as crispra). Different settings were assessed either auto-amplification of well-known strong promoter, weakly activated genes, endogenous genes, in vitro but also in vivo in zebrafish larvae. It provides the reader with a comprehensive view and the versatility of the method.

My comments are mostly aimed towards defining better the limitation of such a technology. It would strengthen the manuscript to have some of the following points below addressed at least in writing or when possible at the experimental level:

1) Can the authors describe why NARTA signal is so clean (eg really low signal to noise ratio). For instance images in figure 1 show a bright signal at the locus where nascent transcription is occurring. Presumably the immature H2B transcripts remain labelled for a while with std-MCP-tomato and are likely to be present in the nucleus floating around.

2) Have the author assessed the effects of splicing on NARTA efficiency? Does the system perform better in presence of splicing drug that maintain aTF/RNA complex intact? Or on gene that have low splicing as opposed to gene with high splicing events?

3) In most experiments BFP is used a proxy to assess target gene over-expression, especially in the context of endogenous gene. It would be helpful to assess endogenous transcripts using single molecule FISH to correlate NARTA activity with increase in transcription on a per nucleus basis. Designing probes in the 3'UTR of the target gene would enable to establish a ratio of fluorescent intensity std-tomato-MCP/endogenous RNA. This point addresses the heterogeneity of the response in the system, qPCR on bulk RNA from cell population show an overall response, however it is clear that the spread of the data that shows gene activation in some quantification is quite large some time. The smFISH approach could help to assess the level of heterogeneity in the cell response too.

4) It is well established that a large proportion of genes have intronic enhancers, is there any steric hindrance from the NARTA system in case of a selected gene has an active intronic enhancer? Using widely available histone marks CHIP-SEQ datasets from ENCODE the authors should review whether the selected genes for endogenous expression do possess intronic enhancer. If none of these genes do, they should select a few genes with this feature to assess the efficacy of NARTA in that configuration. Possibly the presence of intronic enhancers might be one of the limitation in NARTA technology.

5) Have the authors attempted to use NARTA knocked in to a proximal enhancer? Using GRO-seq dataset to select eRNA, it should be possible to label such a regulatory element with the BFP-tritag system. I don't expect an experimental response to this but it would be useful to have their views on this potential approach.

6) In the zebrafish experiment there is no internal control to assess the amount of DNA injected across the different conditions. The use of a constitutive mCherry expressing vector and the ratio GFP/Cherry would help a lot to demonstrate the relative increase in GFP expression.

Manuscript #NCOMMS-22-21988-T

Response to the Reviewer Comments

Reviewer #1 (Remarks to the Author):

Summary

The authors describe a new eukaryotic gene activation system, called Narta, that relies on recruiting transcription factors to nascent transcribed RNA. The authors use RNA binding proteins (RBPs) fused to transcriptional activation domains. The RBPs are directed to nascent RNAs with either heterologous recruitment sites inserted into the target gene or Cas13-based programmable RNA targeting systems. The authors demonstrate that the system is effective in a wide range of systems and can be enhanced with simultaneous CRISPRa at the corresponding genomic DNA. The strongest activation occurred when heterologous RNA target sites were inserted in an intron, and the effects were more modest when using dCas13 to target unmodified endogenous transcripts. The authors readily acknowledged this issue. Overall, the paper presents a novel new method with potential utility for basic research on transcription and for synthetic gene activation. I recommend publication after the authors address the comments below.

Response: We are grateful to the Reviewer for positively evaluating our study and clearly advising us on how to revise our manuscript. Following the Reviewer's suggestions, we have performed additional experiments and revised our manuscript. Our point-by-point response to the Reviewer's comments are stated below.

Major Comments

1. *I found the direct comparisons between CRISPRa and Narta to be one of the more compelling parts of the paper, and the data in Figure 5 addressed the questions I had from the start of the paper. The authors might consider restructuring their manuscript to put Fig 5 before Fig 4, which in my view was interesting but not nearly as critical for establishing the utility of Narta.*

Response: We thank the Reviewer for the thoughtful suggestion. We have now put Fig. 5 before Fig. 4. In

addition to miniCMV, LMNA, BAG3 and HPDL, we tested three more genes to compare CRISPRa , Narta and combinational use of CRISPRa & Narta. The new results (Figure R1) further confirmed our main conclusion in the manuscript and were presented in Supplementary Fig. 12.

Figure R1. Comparison and combinatory use of CRISPRa and Narta for gene activation.

2. The authors provide extensive details about transcriptional dynamics, including burst times and amplitudes. The authors should elaborate on why this information is significant and how it can be interpreted. The authors noted on line 147: "These results suggest that Narta activation induced the production of more H2B transcripts, which was further confirmed by qRT-PCR. Consistent with transcriptional activation, the expression of H2B-BFP

was dramatically increased by 7.9-fold upon Narta treatment (Fig. 1e).” Why do we need any information about the bursting patterns if the point is that more transcripts are produced? Is there some unstated significance to the specific changes in burst dynamics? Does it matter that burst duration increased while pauses stayed the same? What would it mean if burst duration was unchanged and pause lengths decreased?

Response: We thank the Reviewer for these constructive comments. Direct visualization of the dynamics of discontinuous transcription provides a real-time readout of gene activity (1-2). In principle, the levels of gene activity can be regulated by modulating the duration, amplitude, or frequency of individual bursts. Therefore, these transcriptional bursting features have been generally quantified to define gene activity using MS2/MCP-FP systems in numerous previous studies (3-5). A previous study revealed that transcription factor (TF) concentration could modulate bursting parameters (burst duration and burst frequency), which thus offered a tunable regulatory range for individual genes (6). Moreover, a recent work indicated that TF/p300 clustering modulated transcriptional bursting kinetics. Longer burst duration and higher burst amplitude were observed upon TF/p300 cocondensation (7). Narta was designed to concentrate more TFs in the transcription center. Thus, we explored whether Narta could modulate transcriptional bursting kinetics. We have now added additional interpretations in both Result and Discussion sections:

Lines 102-106 (Result section)

We first examined whether nascent RNA-guided activators can induce transcriptional activation. In our system, MS2/stdMCP-tdTomato allows quantitative analysis of transcriptional bursting kinetics in real time. TF is known to be a key determinant for modulating transcriptional bursts. Upon TF activation, longer burst durations and higher burst amplitude were observed in human cells^{24, 42}. Thus, we explored whether Narta could activate transcription by altering transcriptional bursts.

Lines 384-390 (Discussion section)

Previous studies have suggested that TF concentration regulates transcriptional bursting kinetics^{24, 42, 75}. Moreover, strong enhancers drive bursts at a higher frequency than weak enhancers, while SEs exhibit similar bursting patterns to strong enhancers, showing relatively longer burst duration and higher burst amplitude^{21, 76}. Following Narta activation, the transcriptional bursting dynamics of both miniCMV and H2B promoters were altered, displaying similar burst characteristics to those of the strong enhancers. Although Narta is an artificial manipulation, its regulatory effect on genes suggests that TF

is a crucial contributor to transcriptional bursting kinetics for adjusting gene activity.

Reference:

1. Tunnacliffe, E. & Chubb, J.R. What Is a Transcriptional Burst? *Trends Genet* **36**, 288-297 (2020).
2. Nicolas, D., Phillips, N.E. & Naef, F. What shapes eukaryotic transcriptional bursting? *Mol Biosyst* **13**, 1280-1290 (2017).
3. Corrigan, A.M., Tunnacliffe, E., Cannon, D. & Chubb, J.R. A continuum model of transcriptional bursting. *Elife* **5** (2016).
4. Lee, C., Shin, H. & Kimble, J. Dynamics of Notch-Dependent Transcriptional Bursting in Its Native Context. *Dev Cell* **50**, 426-435 e424 (2019).
5. Xu, H. et al. TriTag: an integrative tool to correlate chromatin dynamics and gene expression in living cells. *Nucleic Acids Res* **48**, 13013-13014 (2020).
6. Senecal, A. et al. Transcription factors modulate c-Fos transcriptional bursts. *Cell Rep* **8**, 75-83 (2014).
7. Ma, L. et al. Co-condensation between transcription factor and coactivator p300 modulates transcriptional bursting kinetics. *Mol Cell* **81**, 1682-1697 e1687 (2021).

Minor comments

1. Line 27: *"Moreover, Narta provides better activation potency of some expressed genes than CRISPRa and, when used in combination with CRISPRa, has a synergistic effect on gene activation." Are the effects truly synergistic - more than would be expected based on the sum of the individual effects? The discussion in the main text starting on line 294 is careful on this point, but the abstract (line 27) and the Fig 5 legend (line 871) imply synergy. Enhanced activation above either individual mechanisms is still valuable even if the effects are not strictly synergistic. Minor wording changes would be sufficient to address this point.*

Response: We thank the Reviewer for pointing this out. Indeed, it is not clear currently whether CRISPRa and Narta together play a synergistic effect on gene activation. We have modified the description in the Abstract and Fig. 5 legend as follows:

Lines 28-29 (Abstract)

"Moreover, Narta provides better activation potency of some expressed genes than CRISPRa and, when used in combination with CRISPRa, has an enhancing effect on gene activation."

Line 908-909 (Figure legend)

“a, Schematic of different activators and the potential enhancing effect on gene activation between CRISPRa and Narta.”

2. On line 44, the authors suggest that new approaches are needed to upregulate highly-expressed genes. Have the authors actually demonstrated that the Narta method is better than other gene activation methods at highly-expressed genes? If so, please elaborate. If not, this point may not be necessary. Improved gene activation methods are useful and activating highly-expressed genes may not be the most compelling application.

Response: We thank the Reviewer for pointing this out and we agree that we do not need to emphasize highly-expressed genes. We have modified the introduction as follows:

Lines 41-46

One major concern about the use of CRISPRa is that gene activation efficiency is highly dependent on the sgRNA design and selection. Moreover, the moderate levels of gene activation by CRISPRa have limited its broad applications¹⁴. Therefore, it is still possible to develop a more reliable approach to activate gene expression with wider dynamic ranges. This could be highly desirable for some biological processes, such as the direct conversion of cell types and industrial applications^{15, 16}.

3. The authors used MCP-MS2 to recruit activators and for imaging nascent RNA production, which means the activators and the fluorescent probes are competing for the same RNA targets. This strategy seems less than ideal. I certainly would not suggest that the authors repeat all of their experiments with orthogonal strategies, but the authors could acknowledge a potential concern here and discuss strategies to build orthogonal activation and reporting systems.

Response: We thank the Reviewer for this comment. To assess whether Narta activates gene expression by modulating transcriptional bursting kinetics, we used MCP-MS2 to recruit activators for gene activation and fluorescent proteins for nascent RNA imaging. In our study, this system was only applied for testing miniCMV and H2B. We tested the Narta activation of other exogenous and endogenous genes using cells without stdMCP-tdTomato expression. Following the Reviewer’s suggestion, we have discussed the potential concern in the Result section:

Lines 94-99

It is worth noting that the simultaneous use of stdMCP-tdTomato and stdMCP-PH to image and manipulate gene expression in the same cells may reduce the sensitivity of imaging and the efficiency of gene activation. An orthogonal activation or RNA reporter system would be an ideal design. However, our BFP^{TriTag} harbors 12 copies of the MS2 sequence, which is still worth testing. When only the level of single-cell protein expression needs to be quantified for assessing Narta activation in the subsequent experiments, we used cells without stdMCP-tdTomato expression.

4. Starting on Line 102 and throughout the manuscript, the authors report values to an absurdly large number of significant figures. What is the significance of a burst duration of precisely 16.71 minutes? Is it meaningfully different than 17 minutes? I recommend that the authors re-evaluate the precision with which they report all of the values in the manuscript.

Response: We thank the Reviewer for the insightful comments. We agree that there is no meaningful difference between 16.71 min and 17 min to report burst durations. Following the Reviewer's suggestion, we have re-evaluated the precision of all values reported in this manuscript and revised them accordingly.

5. On line 140, the authors write: "Next, to monitor the activation of an endogenous gene, we inserted TriTag into the C-terminus of human H2B..." Opinions could differ here, but once the genome needs to be edited to include a heterologous sequence I would no longer consider the gene to be "endogenous". A tool to activate endogenous genes should ideally target unmodified endogenous genes, which the authors address later in the manuscript. Once you have decided to edit the genome, why not simply introduce a strong inducible promoter?

Response: We thank the Reviewer for pointing this out. We now used "endogenous promoters" instead of "endogenous genes". We hope the Reviewer will find this change acceptable.

6. Line 184: "Additionally, the DNA size of 12-copy design is much smaller than that of 24-copy, making it more suitable for CRISPR-mediated knockin." Does the size difference between 12 and 24 MS2 repeats really make a large difference for knockin efficiency? I'd generally agree working with 12x MS2 is easier than 24x MS2, but more because larger numbers of repeats can be harder to clone and genetically manipulate.

Response: We appreciate the Reviewer's advice. BFP Tag harbors 0x, 6x, 12x or 24x MS2 is 800, 1128, 1498 and 2167 bp, respectively. We quantified the knockin efficiencies of these tags at LMNA or H2B loci and found that the knockin efficiency was indeed affected by the size of inserts. However, there is only a mild difference between 12x and 24x MS2 tags (Figure R2). Following the Reviewer's suggestion, we have added additional interpretation in the Result section:

Lines 196-199

Additionally, the DNA size of 12-copy design is smaller than that of 24-copy, possibly making it more suitable for molecular cloning of the donor plasmid. Additionally, the smaller size of NarTag may facilitate higher successful rates of CRISPR-mediated knockin (Supplementary Fig. 8d).

Figure R2. Knock-in rates of different NarTags at LMNA or H2B loci.

7. Line 186: The authors compare Narta recruitment sites in UTRs and introns, and observe that intron recruitment gives stronger activation. Can the authors speculate why introns and UTRs behave so differently? Is this observation telling us anything useful about the underlying mechanism?

9. Line 355: "Our results reveal that tethering activators to the intron of nascent pre-mRNA is much more efficient than its UTR region, suggesting that the concentration of activator-bound RNAs (probably spliced introns) may play a critical role in Narta." This statement is unclear. Why is the concentration of activator-bound RNAs different if the same number of recruitment sites are displayed in a UTR or an intron?

Response: We appreciate all the insightful comments in # 7 and 9. First, we further confirmed the conclusion

by measuring the RNA abundance by qPCR. Consistent with single-cell protein quantifications, MS2 in the intron did induce higher transcriptional activation (Figure R3). TFs recruited by UTR may be transported rapidly out of the transcription center along with mature RNAs. However, TFs bound to spliced introns may act directly on the transcriptional activation in the transcription center. Spliced intronic RNA may form stable condensates with its binding partner TFs and downstream co-activators, greatly increasing the local concentration of TFs and coactivators. However, this is just speculation. It would be interesting to explore the underlying mechanism of Narta activation in the future. We have discussed this a little bit in the Discussion section:

Lines 372-377

Our results reveal that tethering activators to the intron of nascent RNAs is much more efficient than its UTR region, suggesting that the activator-bound intronic RNAs (probably spliced introns) may play a critical role in Narta. We speculate that spliced intronic RNA-TFs may be retained in the transcription center and function to mediate transcriptional activation. The underlying mechanisms of Narta activation need to be further investigated.

Figure R3. Narta activation via intronic MS2 is more effective than UTR MS2.

8. On line 259, the authors discuss using Narta to overexpress fluorescent reporters for imaging. There are

discussions throughout the cell biology literature about whether the behavior and localization of overexpressed proteins accurately reflects their behavior at the endogenous expression level. The authors are encouraged to acknowledge these concerns.

Response: We thank the Reviewer for this advice and we agree that the concern of overexpressing proteins should be acknowledged. We have added the following information in the Discussion section:

Lines 406-408

Notably, overexpressed proteins may lead to artifacts, including mislocalizations and protein aggregation⁸⁰. Because Narta is a tunable activation tool, different levels of gene activation can be considered when implementing Narta.

Reviewer #2 (Remarks to the Author):

In this study, Liang and colleagues report the development of a novel molecular approach that drives gene over-expression in a tunable manner. As opposed to current existing methods that are based on synthetic activators /DNA interactions, this novel method is based on synthetic activators/nascent RNA interactions.

The authors have to be acknowledged for their creativity and their ability to take advantage of the MS2/MCP RNA reporter system, which is a technology traditionally used to study transcription kinetics in real time.

The work reported here is quite mature and advanced, the science is sound, with overall a lot of carefully designed controlled experiments and appropriate quantitative measurements. The method seems well established, robust, efficient and reproducible. Overall the data presented are fairly convincing and this original method is likely to provide a great adjunct to the current toolbox in the field.

It is great to see that the authors spent a lot of efforts to not only characterize in depth the efficacy of their system but also to benchmark it against other technology (such as crispra).

Different settings were assessed either auto-amplification of well-known strong promoter, weakly activated genes, endogenous genes, in vitro but also in vivo in zebrafish larvae. It provides the reader with a comprehensive view and the versatility of the method.

My comments are mostly aimed towards defining better the limitation of such a technology. It would strengthen

the manuscript to have some of the following points below addressed at least in writing or when possible at the experimental level:

Response: We thank the Reviewer for appreciating the novelty and the impact of the Narta technology.

Following the Reviewer's constructive suggestions, we have performed additional experiments and provided new data to improve the quality of our manuscript. Our point-by-point responses to the reviewer's comments are stated below.

1) Can the authors describe why NARTA signal is so clean (eg really low signal to noise ratio). For instance images in figure 1 show a bright signal at the locus where nascent transcription is occurring. Presumably the immature H2B transcripts remain labelled for a while with std-MCP-tomato and are likely to be present in the nucleus floating around.

Response: We thank the Reviewer for raising this interesting point. In Figure 1, stdMCP-tdTomato binds newly produced nascent RNAs, so bright stdMCP-tdTomato spots represent abundant RNAs in the transcription center which has been validated in our previous studies (8). stdMCP-tdTomato spots became brighter (with a high signal-to-noise ratio) upon Narta activation, indicating that more nascent RNAs were being produced. Interestingly, we did observe that in many cases nascent RNAs were always enriched at transcription sites (this study and our published or unpublished work). Even though more nascent RNAs were accumulated due to the block of splicing by small drugs, they remain enriched at the production sites. We speculate that there may be mechanisms that retain nascent RNAs in transcription centers to ensure that they are fully transcribed and processed. On the other hand, RNAs that diffuse into the nucleus should appear as single RNAs, which may not be able to be detected by stdMCP-tdTomato due to the pool signal-to-noise ratio.

Reference:

8. Xu, H. et al. TriTag: an integrative tool to correlate chromatin dynamics and gene expression in living cells. *Nucleic Acids Res* **48**, 13013-13014 (2020).

2) Have the author assessed the effects of splicing on NARTA efficiency? Does the system perform better in

presence of splicing drug that maintain aTF/RNA complex intact? Or on gene that have low splicing as opposed to gene with high splicing events?

Response: These are excellent points. Inspired by the Reviewer, we performed some experiments to address the questions. Isoginkgetin is a pre-mRNA splicing inhibitor. We first sought to assess Narta activation by quantifying protein expression levels in the absence or presence of Isoginkgetin. However, Isoginkgetin treatment for 24h caused cell death severely. We then performed smFISH to quantify RNA abundance following Narta activation for 24h and Isoginkgetin treatment for the last 12h during Narta activation. We found that the block of pre-mRNA splicing led to the significant accumulation of mRNA in the nucleus (Figure R4). Thus, it seems that the presence of splicing drugs is not a practical approach to improve Narta efficiency.

It would be interesting to test whether Narta acts differently on genes with low or high splicing events. Our concern is that the degree of activation of different genes can be influenced by multiple factors, making it difficult to draw reliable conclusions from comparing different genes. Perhaps we could make different NarTags with introns that are spliced at fast or slow kinetics, use them to tag the same gene, and then compare them. Unfortunately, we are unable to provide the data here due to time constraints during revision. However, it is definitely worth trying in the future.

Figure R4. The block of mRNA splicing inhibits mRNA being transported out of nuclei.

3) In most experiments BFP is used a proxy to assess target gene over-expression, especially in the context of endogenous gene. It would be helpful to assess endogenous transcripts using single molecule FISH to correlate

NARTA activity with increase in transcription on a per nucleus basis. Designing probes in the 3'UTR of the target gene would enable to establish a ratio of fluorescent intensity std-tomato-MCP/endogenous RNA. This point addresses the heterogeneity of the response in the system, qPCR on bulk RNA from cell population show an overall response, however it is clear that the spread of the data that shows gene activation in some quantification is quite large some time. The smFISH approach could help to assess the level of heterogeneity in the cell response too.

Response: We thank the Reviewer for the constructive comments. Following the suggestion, we have performed the smFISH approach to quantify transcripts in single cells. We designed smFISH probes which specifically recognize BFP exons. Thus, all endogenous genes tagged with BFP^{NarTag} could be activated by Narta and the transcripts could be quantitatively assessed by smFISH (Figure R5). The new results have been added to the Result section:

Lines 174-176

Moreover, mRNA abundance of target genes was dramatically increased measured by qRT-PCR and single-molecule fluorescence in situ hybridization (smFISH), demonstrating that Narta can robustly modulate the transcriptional level at endogenous loci (Fig. 1i and Supplementary Fig. 5).

Figure R5. Detection of Narta activation by smFISH.

4) It is well established that a large proportion of genes have intronic enhancers, is there any steric hindrance from the NARTA system in case of a selected gene has an active intronic enhancer? Using widely available histone marks CHIP-SEQ datasets from ENCODE the authors should review whether the selected genes for endogenous expression do possess intronic enhancer. If none of these genes do, they should select a few genes with this feature to assess the efficacy of NARTA in that configuration. Possibly the presence of intronic

enhancers might be one of the limitation in NARTA technology.

Response: We thank the Reviewer for the insightful comments and suggestions. As suggested, we used H3K27ac CHIP-SEQ datasets from ENCODE to detect whether our target genes contain intronic enhancers. In addition, we confirmed candidates in the enhancer dataset established by Richard Young's group (9). LMNA, HSPB1, HSPB8, BAG3, CYB5B, VAPB and ACTB were found to contain intronic enhancers. Our results showed that these genes could be effectively activated by Narta. Thus, the Narta system appears to have no steric hindrance to the action of intronic enhancers. This seems explainable. Narta recruits abundant TFs to the transcription center through RNA-TF interaction, while intronic enhancers enrich TFs through DNA-TF binding.

Reference

9. Hnisz, D. et al. Super-enhancers in the control of cell identity and disease. *Cell* **155**, 934-947 (2013).

5) Have the authors attempted to use NARTA knocked in to a proximal enhancer? Using GRO-seq dataset to select eRNA, it should be possible to label such a regulatory element with the BFP-tritag system. I don't expect an experimental response to this but it would be useful to have their views on this potential approach.

Response: We thank the Reviewer for the thoughtful comments. eRNAs have been suggested to bring transcription activators to the promoters of neighboring protein-coding genes (10,11). For example, eRNAs at the enhancer can trap transcription factor YY1 and enhance their binding to enhancers (12). Therefore, if we can tag eRNA with NarTag, it may further enhance the effect of eRNA on transcriptional activation. Due to the time constraints of this revision, we were unable to do this experiment. However, it is definitely worth testing this idea in the future.

Reference:

10. Sartorelli, V. & Lauberth, S.M. Enhancer RNAs are an important regulatory layer of the epigenome. *Nat Struct Mol Biol* **27**, 521-528 (2020).

11. Mao, R. et al. Enhancer RNAs: a missing regulatory layer in gene transcription. *Sci China Life Sci* **62**, 905-912 (2019).

12. Sigova, A.A. et al. Transcription factor trapping by RNA in gene regulatory elements. *Science* **350**, 978-981 (2015).

6) In the zebrafish experiment there is no internal control to assess the amount of DNA injected across the different conditions. The use of a constitutive mCherry expressing vector and the ratio GFP/Cherry would help a lot to demonstrate the relative increase in GFP expression.

Response: We thank the reviewer for the excellent suggestion. Following the suggestion, we have used a constitutive mCherry expression vector as the internal control to assess the amount of DNA injected across the different conditions (Figure R6). The following new results have been added to the Result section.

Lines 141-146

We microinjected the mixture of CMV-mCherry & CMV-GFP, CMV-mCherry & CMV-GFP^{Fish_NarTag} & CMV-stdMCP-PH or CMV-mCherry & CMV-GFP^{Fish_NarTag} & CMV-stdPCP-PH into zebrafish embryos of 1-cell stage. We then assessed GFP and mCherry expression by fluorescent imaging when the embryos were developed into the stage of 80% to 90% epiboly (Supplementary Fig.3b). The constitutive mCherry expression vector was used as the internal control to assess the amount of DNA injected across different conditions.

Figure R6. Narta activates the exogenous reporter gene in Zebrafish embryos.

REVIEWERS' COMMENTS

Reviewer #1 (Remarks to the Author):

Summary

The authors have satisfactorily addressed the reviewer comments and the manuscript should be published without further review after the authors consider the comments below.

Comments

1. In response to one of my previous comments, the authors introduced the following sentence to the introduction (line 41): "One major concern about the use of CRISPRa is that gene activation efficiency is highly dependent on the sgRNA design and selection. Moreover, the moderate levels of gene activation by CRISPRa have limited its broad applications." This statement is true, but the new method described in this paper doesn't really resolve these challenges and is almost certainly subject to similar issues. The authors could consider scaling this statement back. One potential alternative perspective is that different methods for gene activation might provide access to genes that are inaccessible to current CRISPRa methods.

2. On line 104, the authors write: "TF is known to be a key determinant for modulating transcriptional bursts." I'm not really sure what the authors mean here. Is this a generic statement that transcription factors modulate transcriptional bursting? Or are they saying that different transcription factors have different effects on transcriptional bursts? Please consider clarifying this point.

3. On line 389, the authors write: "Although Narta is an artificial manipulation, its regulatory effect on genes suggests that TF is a crucial contributor to transcriptional bursting kinetics for adjusting gene activity." Similar to the above comment, this statement is a bit vague. What does it mean to say that a transcription factor is a contributor to bursting kinetics? If I understand the point correctly, it might be more straightforward to write something like: "Although Narta is an artificial manipulation, its regulatory effect on genes suggests that this system modulates transcriptional bursting kinetics in a similar manner to the behavior of endogenous transcription factors."

Reviewer #2 (Remarks to the Author):

The authors has satisfactorily addressed my comments and suggestions. They made the special efforts to generate better controlled experiments using smFISH to assess heterogeneity of the NARTA technology or made double transgenic reporter zebrafish model systems.

I have no further request, this technology will hopefully benefit the field of gene regulation with direct application in bio-engineering and biomedical studies.

Manuscript #NCOMMS-22-21988A

Response to the Reviewer Comments

Reviewer #1 (Remarks to the Author):

Summary

The authors have satisfactorily addressed the reviewer comments and the manuscript should be published without further review after the authors consider the comments below.

Response: We thank the Reviewer for their appreciation of Narta and for the further comments to improve the manuscript.

Comments

1. *In response to one of my previous comments, the authors introduced the following sentence to the introduction (line 41): “One major concern about the use of CRISPRa is that gene activation efficiency is highly dependent on the sgRNA design and selection. Moreover, the moderate levels of gene activation by CRISPRa have limited its broad applications.” This statement is true, but the new method described in this paper doesn’t really resolve these challenges and is almost certainly subject to similar issues. The authors could consider scaling this statement back. One potential alternative perspective is that different methods for gene activation might provide access to genes that are inaccessible to current CRISPRa methods.*

Response: We thank the Reviewer for the thoughtful comments. Following the Reviewer’s suggestion, we have modified the introduction as follows:

Lines 41-46

Based on this principle, the recently developed DNA-targeting platform, CRISPR-Cas9, has enabled the recruitment of artificial transcription factors (aTFs) to any specific genomic site to induce endogenous gene activation, termed CRISPR activation (CRISPRa)⁹⁻¹³. The ability of CRISPRa to activate target genes by using single sgRNAs enables genome-wide transcriptional activation screens^{14, 15}. However, the use of multiple sgRNAs tiled across the target gene promoter can

significantly improve CRISPRa efficiency, suggesting that recruiting as many TFs as possible may be crucial for activating gene expression with wider dynamic ranges^{9-11, 16}. Thus, a new gene activation strategy based on this principle may effectively activate genes that are inaccessible to current CRISPRa methods. This could be highly desirable for some biological processes, such as the direct conversion of cell types and industrial applications^{17, 18}.

2. On line 104, the authors write: "TF is known to be a key determinant for modulating transcriptional bursts." I'm not really sure what the authors mean here. Is this a generic statement that transcription factors modulate transcriptional bursting? Or are they saying that different transcription factors have different effects on transcriptional bursts? Please consider clarifying this point.

Response: We thank the Reviewer for pointing this out. It would be more appropriate to introduce the specific study that suggests that TF concentration modulates transcriptional bursts. We have modified the description as follows:

Lines 41-46

We first examined whether nascent RNA-guided activators can induce transcriptional activation. In our system, MS2/stdMCP-tdTomato allows quantitative analysis of transcriptional bursting kinetics in real time. A number of studies have established the link of TF and transcriptional bursts^{26, 43, 44}. It was previously suggested that TF concentration can modulate the burst frequency⁴³. Thus, we explored whether Narta could activate transcription by altering transcriptional bursts.

3. On line 389, the authors write: "Although Narta is an artificial manipulation, its regulatory effect on genes suggests that TF is a crucial contributor to transcriptional bursting kinetics for adjusting gene activity." Similar to the above comment, this statement is a bit vague. What does it mean to say that a transcription factor is a contributor to bursting kinetics? If I understand the point correctly, it might be more straightforward to write something like: "Although Narta is an artificial manipulation, its regulatory effect on genes suggests that this system modulates transcriptional bursting kinetics in a similar manner to the behavior of endogenous transcription factors."

Response: We thank the Reviewer for this suggestion. We have adjusted the description as recommended.

Lines 388-392

Following Narta activation, the transcriptional bursting dynamics of both miniCMV and H2B promoters were altered, displaying similar burst characteristics to those of the strong enhancers. Although Narta is an artificial manipulation, its regulatory effect on genes suggests that this system modulates transcriptional bursting kinetics in a similar manner to the behavior of endogenous transcription factors.

Reviewer #2 (Remarks to the Author):

The authors has satisfactorily addressed my comments and suggestions. They made the special efforts to generate better controlled experiments using smFISH to assess heterogeneity of the NARTA technology or made double transgenic reporter zebrafish model systems.

I have no further request, this technology will hopefully benefit the field of gene regulation with direct application in bio-engineering and biomedical studies.

Response: We thank the Reviewer again for their appreciation of our new gene activation method and for their helpful comments and suggestions to improve the manuscript.